# Bridging Reasoning to Learning: Unmasking Illusions using Complexity Out-of-Distribution Generalization

**Mahdi Samiei**                                                           *mm.samiei@sharif.edu*
*Department of Computer Engineering*
*Sharif University of Technology*

**Arash Marioriyad**                                                  *arashmarioriyad@gmail.com*
*Department of Computer Engineering*
*Sharif University of Technology*

**Arman Tahmasebi-Zadeh**                                 *arman.tahmasebi345@sharif.edu*
*Department of Computer Engineering*
*Sharif University of Technology*

**Mohamadreza Fereydooni**                                *mrezafereydooni@gmail.com*
*Department of Computer Engineering*
*Sharif University of Technology*

**Mahdi Ghaznavai**                                          *mahdi.ghaznavi@ce.sharif.edu*
*Department of Computer Engineering*
*Sharif University of Technology*

**Mahdieh Soleymani Baghshah**                              *soleymani@sharif.edu*
*Department of Computer Engineering*
*Sharif University of Technology*

**Reviewed on OpenReview:** *https://openreview.net/forum?id=07fh13gWs0*

## Abstract

Recent progress has pushed AI frontiers from pattern-recognition tasks toward problems that require step-by-step, System-2-style reasoning, especially with large language models. Yet, unlike learning, where generalization and out-of-distribution (OoD) evaluation concepts are well formalized, there is no clear, consistent definition or metric for "reasoning ability." We propose Complexity Out-of-Distribution (Complexity OoD) generalization as a framework and problem setting to measure reasoning. A model exhibits Complexity OoD generalization when it maintains performance on test instances whose minimal required solution complexity, either representational (richer solution structure) or computational (more reasoning steps/program length), exceeds that of all training examples. We formalize complexity via solution description Kolmogorov complexity and operational proxies (e.g., object/relation counts; reasoning-step counts), clarifying how Complexity OoD differs from length and compositional OoD. This lens unifies learning and reasoning: many cases solvable with System-1-like processing at low complexity become System-2-like under complexity pressure, while System-2 can be viewed as generalization over solution structures. We translate this perspective into practice with recommendations for operationalizing Complexity OoD across the stack: incorporating complexity into benchmark and evaluation metric design, rethinking supervision to target solution traces (from final outcomes to process-level feedback and RL/search), seeking and designing inductive biases for Complexity-OoD generalization, addressing learning-to-reason spillovers such as spurious shortcuts, semantic robustness, catastrophic forgetting, and step-wise calibration. In light of recent controver-

sies over LLM reasoning, we put the problem on firm footing: treat reasoning as Complexity OoD, enabling rigorous evaluation and more systematic research.

# 1 Introduction

What do the concepts of intelligence, thinking, and specifically reasoning mean, and by what criteria can we confidently assert that one agent possesses superior reasoning ability compared to another? In parallel with these philosophical inquiries, cognitive science introduces the distinction between System-1 and System-2 thinking Kahneman (2011); Stanovich & West (2000). System-1 processes are rapid, intuitive, and rely heavily on pattern recognition. Many current AI achievements, particularly in areas like computer vision and NLP, demonstrate strong System-1-like capabilities by excelling at pattern recognition tasks Goyal & Bengio (2022); Krizhevsky et al. (2012); Vaswani et al. (2017). The evaluation of AI models trained for these tasks has traditionally centered on their ability to generalize to unseen data. This is typically measured by performance on a held-out test set drawn from the same underlying distribution as the training data (often termed in-distribution generalization) Goodfellow et al. (2016); Zhang et al. (2017). However, a growing recognition in the field highlights the crucial importance of out-of-distribution (OoD) generalization Geirhos et al. (2020b); Recht et al. (2019); Hendrycks et al. (2020); Arjovsky et al. (2019); Gulrajani & Lopez-Paz (2021). This more challenging form of generalization assesses a model's robustness and true understanding by evaluating its performance on data that significantly deviates from its training distribution. Consequently, modern benchmarks and evaluation methodologies are increasingly incorporating OoD splits alongside traditional test sets to better gauge a model's true learning capabilities and its ability to handle novel, real-world scenarios Koh et al. (2020); Gagnon-Audet et al. (2022); Sagawa et al. (2020); Taori et al. (2020).

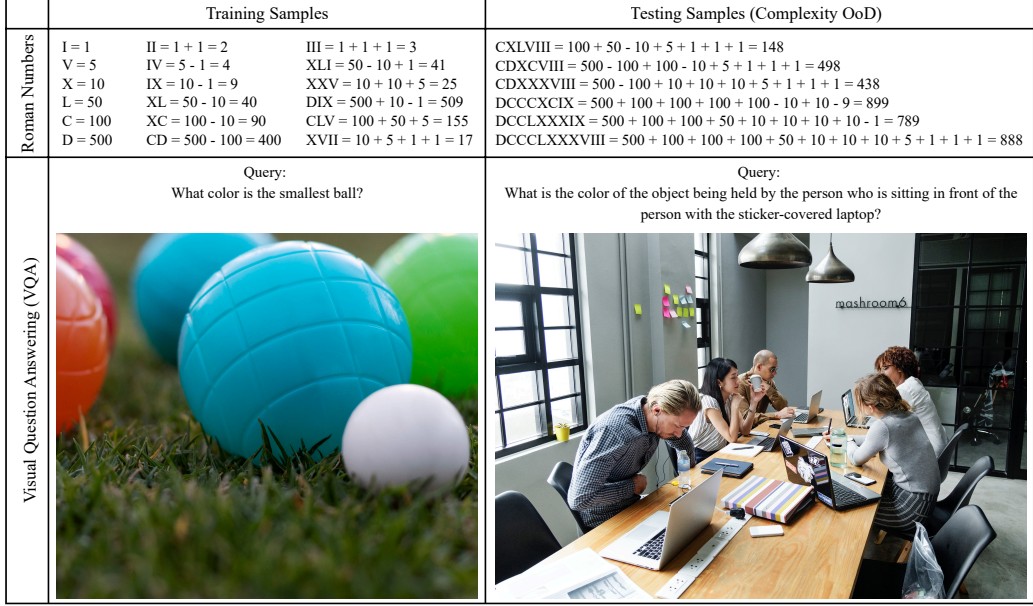

Figure 1: Complexity out-of-distribution generalization evaluates whether models trained on problems whose solutions require few, shallow steps generalize to problems whose solutions demand substantially more steps and deeper composition. Two instantiations are shown. Top-Roman numerals: training solutions involve short additive/subtractive decompositions; test solutions require many more operations to expand more complex numerals. Bottom-Visual Question Answering: training features few-hop questions in a simple scene; testing uses relational, multi-hop questions in a busier scene with more entities. The shift is in solution complexity (and, for VQA, scene clutter), not in domain.

Alongside these rapid System-1 processes, cognitive science also identifies System-2 thinking. These are characterized by slow, deliberate, and effortful operations, involving analytical thought, complex problem-

solving, and crucially, the ability to construct multi-step solutions Kahneman (2011); Stanovich (2011). These System-2 processes directly correspond to the challenging reasoning tasks that have recently gained significant prominence in the field of artificial intelligence, particularly with the rise of large language models (LLMs) Goyal & Bengio (2022); Li et al. (2025); Wei et al. (2022b). However, unlike the well-defined metrics of in-distribution and out-of-distribution generalization that evaluate System-1 tasks, there currently lacks a clear and transparent framework for consistently defining and measuring generalization for System-2 reasoning abilities Mondorf & Plank (2024); Raji et al. (2022).

In recent years, a proliferation of benchmarks has been introduced, aimed at quantifying the reasoning capabilities of Large Language Models (LLMs) Cobbe et al. (2021); Gao et al.; Hendrycks et al. (2021); bench authors (2023); Huang & Chang (2023). While these models have demonstrated truly impressive performance on many of these tasks, the early benchmarks exhibited several notable limitations. Firstly, their evaluation was often solely predicated on the correctness of the final answer, neglecting the actual reasoning process that led to it Lightman et al. (2023). Secondly, and perhaps more critically, while often focusing on narrow domains like mathematics and programming, these benchmarks inadvertently limited their scope, failing to capture the broader, fundamental nature of reasoning itself beyond domain-specific problem-solving Wei et al. (2022b). Finally, by failing to account for the underlying distribution of problem instances, they provided an insufficient fine-grained assessment of model performance and inherent limitations, making it difficult to precisely diagnose where and why models struggled Shojaee et al. (2025); Taori et al. (2023). More recently, systematic investigations into LLMs performance across varying problem difficulties have revealed a critical disconnect: performance on more challenging instances often does not scale proportionally with, or meet the expectations set by, their performance on simpler ones. This observation suggests that models' strong performance on simpler examples might be artificially boosted by data exposure or contamination, blurring the line between genuine reasoning and mere memorization Shojaee et al. (2025); Zhou et al. (2025b); Sun et al. (2025c); Mirzadeh et al. (2025); Golchin et al. (2023). Crucially, this evaluation approach offers minimal insight into the intricate structure or quality of the reasoning traces themselves, making it difficult to truly understand how models arrive at their answers or the robustness of their internal processes Wang et al. (2023a). Consequently, the fundamental question of how to reliably discern which model possesses superior reasoning ability, in the absence of a universally accepted definition and robust criteria for genuine reasoning, remains a significant and largely unresolved challenge.

Addressing the persistent challenge of measuring reasoning, this work introduces Complexity Out-of-Distribution (Complexity OoD) generalization as a novel conceptual framework. Herein, "reasoning ability" is fundamentally reinterpreted as a model's capacity for this specific type of OoD generalization bench authors (2023). Complexity OoD is formally defined as a scenario where the inherent complexity distribution of test samples significantly surpasses that observed in the training data. Within this framework, 'Complexity' is understood as either the requisite representational capacity or the total number of necessary solution steps for a given problem instance. It is hypothesized that truly superior models are those capable of robustly satisfying this Complexity OoD criterion. The integration of this perspective into evaluation and benchmarking protocols is expected to yield assessments that are markedly more robust against data contamination and offer a more precise, nuanced measure of a model's foundational capabilities Taori et al. (2023). Moreover, this framework elucidates a crucial conceptual interplay between conventionally delineated 'learning' (System-1) and 'reasoning' (System-2) tasks Stanovich (2011). It is argued that numerous tasks that are typically handled via System-1 processing, when challenged by instances exhibiting Complexity OoD, inherently evolve into problems demanding a System-2 (reasoning-based) approach for successful resolution. Conversely, by analyzing reasoning through the paradigm of Complexity OoD generalization, it is demonstrated that every System-2 solution can, in turn, be construed as an advanced form of 'Learning' and generalization. This unified perspective aims to bridge the long-standing conceptual divide between learning and reasoning, thereby contributing a more comprehensive framework for comprehending intelligence.

The subsequent sections of this paper elaborate on these contributions. First, a more precise definition of Complexity OoD is provided. This includes differentiating the concept from similar settings, such as compositional OoD, and formally defining it by leveraging principles of Kolmogorov complexity. Next, it is demonstrated how considering Complexity OoD can bridge the concepts of System-2 thinking and learning, revealing that the successful mastery of any System-2 processing often inherently relies on the underlying

learning of a System-1-like component. The paper then illustrates that Complexity OoD is not an entirely alien concept within the field; rather, its facets have been observed across various domains, albeit without a unified, overarching perspective. Finally, and most importantly, this work elaborates on the necessary shifts in research methodology and evaluation priorities within the field that arise when assessing models' reasoning abilities through the lens of Complexity OoD.

## 2 Complexity Out of Distribution

### 2.1 Motivation

As previously discussed, cognitive processing can be broadly categorized into two modes: System-1, which is fast and intuitive, and System-2, which is slow and deliberative. This dichotomy is visibly mirrored in the prevailing paradigms of artificial intelligence over the last decade Goyal & Bengio (2022); Lowe (2024). Tasks addressed within the System-1 framework, such as classifying an image with a fine-tuned ResNet or a piece of text with a BERT model, typically employ an architecture of a fixed computational depth to map an input directly to an output. In this System-1 approach, the primary objective is generalization to unseen samples from the same data distribution or, at best, generalization to a distribution that has undergone a statistical shift Vapnik (1998); Quionero-Candela et al. (2009); Hendrycks & Dietterich (2019); Geirhos et al. (2020a).

In contrast, a range of tasks is approached from a System-2 perspective. Examples include solving a mathematical problem with a Large Language Model (LLM) Wei et al. (2022b), answering a complex visual query with a Vision-Language Model (VLM) Liu et al. (2023), or, more abstractly, solving a symbolic regression problem Biggio et al. (2021). In all such cases, "solving" the problem is synonymous with "generating a solution", a coherent sequence of logical sub-steps. The central challenge, therefore, becomes the synthesis of the correct sequence.

The System-2 perspective raises a critical question: what if a model, ostensibly trained with a System-2 approach, merely "memorizes" the simple and short solution paths present in its training data? Such a model's capacity would be confined to generating solutions of low complexity. Consequently, when faced at test time with an instance requiring a solution path that exceeds this capacity, the model will fail. From the model's perspective, such an instance is out-of-distribution with respect to the complexity of its solution.

We term this scenario Complexity Out-of-Distribution (Complexity OoD) generalization. It dictates that a System-2-based model must be able to generalize over problem instances whose solution complexity is out-of-distribution relative to all training examples. To overcome Complexity OoD, a model must possess a crucial, dynamic capability: the ability to generate a solution of any required complexity on the fly. In other words, during inference, the model must be able to dynamically extend its reasoning process, creating a solution path more complex than any it has seen before.

It is critical to note that the Complexity OoD challenge cannot be resolved merely by scaling training data, as one can always conceive of a test instance with a solution complexity greater than any found in the training set. Consequently, achieving Complexity OoD generalization requires the incorporation of appropriate inductive biases. We posit that if a model can guarantee Complexity OoD generalization for a given task, it can then achieve perfect generalization for any instance of that task.

Finally, the concept of Complexity OoD is not exclusively confined to System-2 approaches. A developer might build a System-1-style model that performs excellently on training examples with limited solution complexity. However, this same model will likely fail when confronted with a test instance that is Complexity OoD, revealing the hidden limitations of its fixed-depth architecture and underscoring the universal importance of this evaluative dimension Hahn (2019); Santoro et al. (2018).

**Examples of Complexity OoD.** Complexity OoD arises whenever test instances require solutions whose minimal complexity (e.g., number of necessary reasoning steps, proof depth, plan length, or description length) substantially exceeds that of training instances, even when surface statistics remain similar. As illustrated in Figure 1, Roman numerals provide a concrete example: The core task involves converting Roman

numeral strings to their decimal equivalents (and conversely), adhering to the standard additive–subtractive rules. Elementary numerals (e.g., I, V, X or even II, IV and XX) can often be processed by System-1 mechanisms, facilitating rapid, intuitive recognition. Conversely, comprehending more intricate numerals like XIX (19), XXIV (24), or LXXXIX (89) mandates the integration of constituent units via a set of compositional rules. This transition moves beyond simple associative recall, requiring System-2 processes to construct systematic solutions by recursively combining previously acquired elements into a coherent representation aligned with the numeral system's structural logic. Another case arises in arithmetic reasoning. Single-digit multiplications (e.g., $3 \times 4$) can often be recalled directly or processed via System-1 pattern-matching. In contrast, larger multiplications (e.g., $47 \times 89$) require the integration of smaller learned operations into a multi-step algorithm, engaging System-2 processes for systematic computation. [1]

The applicability of Complexity OoD extends far beyond these foundational symbolic domains, providing a powerful lens for designing and analyzing benchmarks across diverse high-difficulty tasks. In the realm of visual reasoning and Visual Question Answering (VQA), for example, Complexity OoD can manifest through either increased visual richness or heightened logical demands in the query. A test scene might be significantly more cluttered with objects, attributes, and relations than any training example. Alternatively, the question itself could demand more reasoning hops. For instance, consider Figure 1, a model trained on single-hop questions like "What color is the smallest ball?" could be challenged with a multi-hop query such as "What is the color of the object being held by the person who is sitting in front of the person with the sticker-covered laptop?" Answering this requires a multi-step inferential chain: identify the person with the sticker-covered laptop, determine who is sitting in front of them, detect the object that person is holding, and then report the color of that object. This principle is equally critical in robotics and long-horizon planning. A robot might be trained on tasks requiring short action sequences (e.g., "pick up the blue block and place it on the red block"). A Complexity OoD test would demand a significantly longer and more intricate plan, such as "build a four-block pyramid, which first requires clearing the table by moving all non-block items into the designated box." This requires not just more steps, but also managing sub-goals and interdependent constraints that were absent in the training data. Similarly, in fields like automated theorem proving, a model trained to prove lemmas requiring proofs of a certain depth (e.g., 5–10 inference steps) would face a Complexity OoD challenge when asked to prove a theorem whose shortest proof is an order of magnitude longer. The model must demonstrate an ability to chain inference rules for a duration far exceeding its training experience. The same logic applies to code generation, where a test problem might require programs with greater structural depth, such as more nested functions, intricate recursive patterns, or control flow with deeper nesting, richer branching, and longer dependency chains than any example in the training set. In algorithmic reasoning, this could involve a path-finding model trained on graphs of a certain size being tested on a graph of a similar size but with a significantly larger diameter, forcing the execution of a much longer reasoning sequence. Finally, the concept is highly relevant to narrative and document comprehension. A model may excel at answering questions about short stories where the causal chain is direct and localized. The true test of its reasoning ability, its Complexity OoD performance, comes from processing a long novel and answering a question about a character's motivation that requires synthesizing subtle clues and events scattered across multiple chapters.

In all these cases, the underlying challenge is the same: the model must dynamically construct a solution or reasoning trace that is structurally more complex than any it has been trained on, moving beyond pattern matching to genuine, scalable procedural understanding. Across these domains, the common failure mode is not exposure to unfamiliar tokens or images per se, but the need to execute solutions whose minimal complexity exceeds the training support. Conversely, models equipped with inductive biases for adaptive,

---

[1]For multiplication, LLMs essentially have two distinct approaches: either generating and executing Python code or attempting to perform the calculation internally, without external tools. As highlighted in length generalization studies, a crucial difference emerges: humans, given sufficient attention and working memory, can accurately perform mathematical operations reliably, exhibiting robust length generalization in mathematical reasoning. LLMs, however, do not possess this same guaranteed length generalization, particularly for complex or lengthy mathematical problems when relying solely on internal computation. This is because LLMs, instead of learning the underlying logic of multiplication, a logic inherently generalizable to numbers of any length, primarily learn to mimic the process as observed in their training data. They are, in essence, pattern-matching procedural steps rather than grasping the abstract mathematical principles themselves. This contrasts sharply with human mathematical understanding, which is built upon a foundational grasp of logical structure that ensures generalizability.

iterative computation and external memory/tools tend to generalize more gracefully along this axis Graves et al. (2014); Dehghani et al. (2019b); Veličković & Blundell (2021); Gao et al. (2023); Schick et al. (2023).

**Distinguishing Complexity OoD from Compositionality** In the literature on compositional generalization, two primary out-of-distribution scenarios are commonly discussed: *systematicity* and *productivity* (Hupkes et al., 2020). The performance of models has frequently been evaluated under these conditions (Lake & Baroni, 2018; Hupkes et al., 2020; Loula et al., 2018). *Systematicity* refers to the ability to generalize to novel combinations of known components, even when such specific combinations were absent during training (Hupkes et al., 2020). *Productivity*, by contrast, refers to the ability to generalize to sequences of greater length than those encountered during training (Hupkes et al., 2020). Although *complexity OoD* bears some conceptual similarity to compositional OoD, it represents a fundamentally different perspective. The distinction between complexity OoD and systematicity lies in the scope of complexity within the compositions. In systematicity, the challenge is bounded: models must recombine a limited number of familiar primitives at a specified composition length, not in constructing longer length compositions. By contrast, complexity OoD imposes no such bound; it emphasizes the need to handle solution paths whose complexity may grow arbitrarily, often requiring deeper reasoning chains characteristic of System-2 processes. The difference between complexity OoD and productivity (often referred to as length OoD) is equally crucial. Length generalization focuses on the size of the input or output sequence, without necessarily implying an increase in reasoning demands. Complexity OoD, however, is defined by the growth of the solution path itself, that is, the number of reasoning or computational steps required to connect input to output. Importantly, these two notions can diverge: a long input may be solvable via a trivial, System-1 style operation, while a short input may require intricate, multi-step System-2 reasoning. For example, a long sequence of repeated symbols (e.g., "aaaaa...") might pose a challenge for productivity but is trivial in terms of reasoning complexity, whereas a short logical puzzle can exemplify complexity OoD by demanding deep multi-step inference despite its brevity.

To make this boundary fully concrete, consider evaluating arithmetic expressions. A model trained on short additions can be tested in two distinct ways. A *length OoD* test extends the surface form while keeping the computation flat, e.g. moving from a 5-term sum to a 20-term sum $a_1 + a_2 + \cdots + a_{20}$: the input string is much longer, but the solution is a flat left-to-right accumulation whose per-step computational depth never grows. A *Complexity OoD* test instead increases the depth of the computation while holding length roughly fixed: compare a flat expression $(10 - 1) + (10 - 2) + (10 - 3) + (10 - 4)$ with a deeply nested one $10 - (10 - (10 - (10 - 1)))$. The two have nearly identical token length, operand count, and number of operations, yet the flat expression admits shallow, order-independent evaluation, whereas the nested expression forces sequential, stateful tracking down a computational syntax tree of increasing depth. A model may handle long flat sums perfectly (good length generalization) yet collapse on short but deeply nested expressions (failing Complexity OoD), and vice versa. This dissociation is exactly what separates the two notions: length generalization probes the capacity to sustain processing over a longer sequence, while Complexity OoD probes the capacity to execute a deeper solution. We return to the implications of this distinction for *proxy design* in Section 2.4, where the maximum depth of the computational syntax tree—rather than raw token count—emerges as the faithful complexity proxy in this setting.

**Representational and Computational Complexity OoD** Complexity can be analyzed along two complementary dimensions: the *representational* and the *computational*. *Representational complexity OoD* arises when test samples exhibit richer or more intricate structures than those observed during training. Such samples demand finer-grained descriptions or higher-dimensional representations in order to be accurately reconstructed or discriminated. The second dimension, *computational complexity OoD*, concerns cases in which obtaining the correct solution requires additional reasoning steps compared to the training regime. Here, the challenge lies not in representing the input but in extending the chain of computation, moving beyond shallow System-1 pattern recognition toward adaptive, multi-step System-2 processing. These two dimensions are deeply intertwined. Representational complexity often induces computational complexity: a richer input representation may necessitate longer reasoning paths, while deeper computation may reveal or require more expressive representational structures. Rather than treating them as isolated phenomena, it is crucial to view representational and computational complexity as two sides of the same coin, each shaping and amplifying the other. Consequently, addressing complexity OoD requires integrated solutions.

A successful framework must accommodate *unbounded representational depth*, which refers to the ability to flexibly encode increasingly complex inputs, as well as *adaptive computational depth*, which refers to the capacity to dynamically extend the number of reasoning steps as needed. We argue that achieving robust System-2 solutions hinges on jointly solving both challenges, thereby enabling models to generalize across variable levels of complexity in real-world data.

## 2.2 Formal Definition of Complexity OoD

System-2 reasoning can be understood as the capacity to handle unbounded complexity in both representation and computation. In contrast to System-1 processing, which often relies on rapid pattern matching and shallow templates for frequent inputs, System-2 processing explicitly constructs and manipulates intermediate structure by composing task-relevant primitives. Here, primitives denote functional building blocks at the current abstraction level—such as symbols, operators, predicates, spans, or object slots—which may be realized as distributed, overlapping representations yet are treated as indivisible within the active composition scheme. Conceptually, System-2 competence requires the ability to assemble these primitives into longer computational chains and richer descriptions on demand, that is, to represent increasingly intricate inputs and to allocate progressively deeper computation when required.

Let us assume a vocabulary of primitives $M = \{m_1, m_2, \ldots, m_n\}$. These primitives may serve as representation primitives (analogous to words) or computational primitives (analogous to basic operators). A System-2 solution can then be described as constructing a correct program over $M$. In the representational setting, the program is a structured description (for example, a sentence); in the computational setting, it is an executable procedure (for example, an equation or algorithm). If we further assume access to an oracle that determines whether a given program achieves the goal, then solving reduces to searching for the shortest valid program within the space of programs over M. [2]

To formalize these ideas, we draw on Kolmogorov Complexity (Kolmogorov, 1965; Li et al., 2008). Although uncomputable in practice, it provides a rigorous theoretical lens for distinguishing between representational and computational complexity in System-2 reasoning.

### 2.2.1 Representational Complexity OoD

Let $x$ be an input sample (e.g., an image, a sentence, or a structured object). Its representational complexity is defined as the Kolmogorov Complexity of $x$, denoted $K(x)$:

$$K(x) = \min\{|p| : U(p) = x\}, \tag{1}$$

where $U(p)$ is the output of a universal Turing machine $U$ given program $p$, and $|p|$ is the program's length (e.g., in bits). Intuitively, $K(x)$ measures the shortest description length of $x$. High values of $K(x)$ indicate that $x$ has rich or intricate structure, requiring more expressive representations.

A *representational Complexity OoD* scenario occurs when a test sample $x_{\text{test}}$ requires a description longer than that of any training instance:

$$K(x_{\text{test}}) > \max_{x_{\text{train}} \in D_{\text{train}}} K(x_{\text{train}}). \tag{2}$$

In this case, the model must cope with a representational demand that exceeds its training distribution.

---

[2]In the following sections, because the term "program" may be confusing or misleading, we will instead use the term "solution" However. By solution we mean a possibly unbounded-length, stepwise program and procedure built from semantic primitives, in System-2 processing to arrive at an answer.

### 2.2.2 Computational Complexity OoD

Let $y$ denote the solution corresponding to input $x$. In System-2 processes, mapping $x \rightarrow y$ often requires a multi-step reasoning procedure. We capture the complexity of this procedure via conditional Kolmogorov Complexity, $K(y \mid x)$:

$$K(y \mid x) = \min\{|q| : U(x, q) = y\}, \tag{3}$$

where $q$ is a program that takes $x$ as input (or encodes it internally) and produces $y$ as output, with $|q|$ denoting its length. A high $K(y \mid x)$ implies that solving for $y$ requires a longer or more intricate computation.

A *computational Complexity OoD* scenario occurs when a test pair $(x_{\text{test}}, y_{\text{test}})$ demands a solution program of strictly greater complexity than any training example:

$$K(y_{\text{test}} \mid x_{\text{test}}) > \max_{(x_{\text{train}}, y_{\text{train}}) \in D_{\text{train}}} K(y_{\text{train}} \mid x_{\text{train}}). \tag{4}$$

### 2.3 Proxies of Complexity OoD

Kolmogorov Complexity provides a rigorous theoretical lens for defining representational and computational complexity, but it is uncomputable in practice (Li et al., 2008). To study Complexity OoD empirically, we rely on practical proxies that approximate these abstract notions.

**An Example of Representational Complexity Proxy.** A useful proxy for representational complexity in the visual domain is the number of objects, attributes, and relations present in a scene. For instance, an image of a single isolated object (e.g., a red cube on a plain background) has low representational complexity, whereas a crowded scene containing multiple overlapping entities with interacting attributes (e.g., "three people sitting at a table surrounded by books and food") has significantly higher representational complexity (Johnson et al., 2017). Importantly, this notion differs from length generalization. In natural language processing, productivity or length OoD refers to the growth of sequence length (e.g., more tokens in a sentence). By contrast, in images the input dimensions remain fixed, and complexity increases not by length but by the richness of semantic content within the same spatial grid.

| Overview of GSM8K Benchmark | |
|---|---|
| Sample with 2 Math Operations | Sample with 8 Math Operations |
| Q: Natalia sold clips to 48 of her friends in April, and then she sold half as many clips in May. How many clips did Natalia sell altogether in April and May? | Q: Tina makes $18.00 an hour. If she works more than 8 hours per shift, she is eligible for overtime, which is paid by your hourly wage + 1/2 your hourly wage. If she works 10 hours every day for 5 days, how much money does she make? |
| A:
Natalia sold 48/2 = 24 clips in May.
Natalia sold 48+24 = 72 clips altogether in April and May. | A:
She works 8 hours a day for $18 per hour so she makes 8*18 = $144.00 per 8-hour shift.
She works 10 hours a day and anything over 8 hours is eligible for overtime, so she gets 10-8 = 2 hours of overtime.
Overtime is calculated as time and a half so and she makes $18/hour so her overtime pay is 18*.5 = $9.00.
Her overtime pay is 18+9 = $27.00.
Her base pay is $144.00 per 8-hour shift and she works 5 days and makes 5 * $144 = $720.00.
Her overtime pay is $27.00 per hour and she works 2 hours of overtime per day and makes 27*2 = $54.00 in overtime pay.
2 hours of overtime pay for 5 days means she makes 54*5 = $270.00.
In 5 days her base pay is $720.00 and she makes $270.00 in overtime pay so she makes $720 + $270 = $990.00 |

Figure 2: Two examples from the GSM8K dataset in which the number of mathematical operations required to solve the problem can be considered as a proxy for the complexity of the sample problem.

**An Example of Computational Complexity Proxy** A natural proxy for computational complexity is the length of the reasoning chain required to derive the correct output, as illustrated in Figure 2. Simple

arithmetic such as $2+3$ requires only one step, whereas solving a multi-step algebraic equation or answering a compositional visual reasoning query (e.g., "Is there a cube to the left of the sphere that is larger than the red object?") requires a sequence of intermediate inferences (Merrill et al., 2023; Wei et al., 2022a). Here, the complexity does not arise from longer inputs but from the depth of reasoning steps. This directly reflects the System-2 requirement: solutions must dynamically expand the number of computational steps to accommodate increasingly complex problem instances.

These proxies make Complexity OoD operational: representational complexity emphasizes the growth of informational richness in inputs, while computational complexity emphasizes the depth of reasoning needed to process them.

## 2.4 Scope and Limitations of Complexity OoD Proxies

While the most intuitive applications of Complexity OoD lie in well-structured, goal-directed domains like mathematics or algorithmic reasoning, the core principle extends inherently to less structured domains such as natural language understanding (NLU) and language modeling. In natural language, complexity often manifests as syntactic or semantic depth rather than explicit step-by-step math operations. For example, comprehending a sentence with deep nested structures, such as "The failure of the vote to remove the anti-war representative...", often requires humans to pause, backtrack, and process the text sequentially to maintain state and resolve long-distance dependencies. In such NLU tasks, the depth of the parse tree or the length of syntactic dependencies can serve as valid complexity proxies. At a higher semantic level, the density and depth of metaphors or implicit pragmatics in a text demand greater computational work (semantic disentanglement) to decode. Thus, Complexity OoD can be conceptualized and defined across a wide spectrum of tasks beyond explicit algorithmic benchmarks.

However, selecting an appropriate proxy to measure this complexity is highly non-trivial and comes with inherent limitations. A common pitfall is strictly conflating solution length with solution complexity. A longer output does not necessarily imply greater algorithmic depth. Consider a prompt asking a model to "Print the word 'Hello' 1,000 times." The output token count is exceptionally high, but the underlying generative process is computationally trivial, requiring minimal reasoning depth. This tests a model's productivity, not its reasoning. Conversely, consider a complex logical riddle or a mathematical problem requiring a sudden "Aha!" insight. The final proof might be only two lines long, but discovering it requires navigating a massive combinatorial space. In such cases, using the length of the final generated output as a complexity proxy severely underestimates the true difficulty, as it entirely ignores the implicit search cost and the internal computational work required to arrive at that short solution.

The flat-versus-nested arithmetic contrast introduced in Section 2.1 also illustrates why the choice of proxy matters. One might be tempted to use the total number of tokens (sequence length) as a proxy for complexity. However, recall that the flat expression $(10-1)+(10-2)+(10-3)+(10-4)$ and the nested expression $10-(10-(10-(10-1)))$ have similar token length, yet the nested form requires deep, sequential, stateful tracking representing a fundamentally higher level of algorithmic complexity. Using sequence length as a proxy here would mistakenly frame the evaluation as a *length generalization* problem. The more faithful proxy for Complexity OoD in this context is the maximum depth of the computational syntax tree. Therefore, faithful complexity proxies must be carefully designed to measure the structural and computational bottlenecks of a specific task, rather than relying on superficial attributes like token counts.

Finally, we clarify the relationship between Complexity OoD and curriculum learning (CL), as both invoke a notion of problem difficulty and both rely on proxies to operationalize it. This comparison is instructive precisely because it underscores the intended role of our complexity proxies: they are meant to serve as a diagnostic signal for evaluation, not as a difficulty signal for ordering training. The curriculum learning literature has repeatedly observed that human-defined difficulty measures often fail to align with the optimal learning trajectory of neural networks Bengio et al. (2009); importantly, this critique targets the use of such measures as a training signal, and therefore does not transfer to our setting. We emphasize two distinctions that insulate our framework from this critique. First, curriculum learning is a training strategy (ordering examples from easy to hard), whereas a Complexity OoD proxy is purely an evaluation and diagnostic instrument: we make no claim about training order, only that a model capable of reasoning must generalize

at test time to instances whose solution complexity exceeds the training support. Second, the difficulty in CL is typically subjective, heuristic, or model-centric (what the current model finds hard to fit), whereas our notion of complexity is grounded in the objective computational structure of the task (e.g., minimal solution depth or required number of inference steps), which in System-2 domains such as mathematics and algorithmic reasoning is closely tied to genuine algorithmic cost rather than perceived hardness. Indeed, a Complexity OoD proxy can be seen as a diagnostic for one failure mode that CL itself struggles with: if a model learns to solve easy instances via superficial shortcuts rather than the underlying algorithm, it will collapse precisely on Complexity OoD test instances, exposing the shortcut.

Crucially, acknowledging the challenges and limitations in proxy design does not invalidate the Complexity OoD framework itself. Rather, it underscores the necessity of moving beyond naive, surface-level heuristics, such as raw token length, when evaluating reasoning capabilities. The fact that a poorly chosen proxy can misrepresent a model's ability simply highlights that evaluating true reasoning demands metrics that faithfully capture the underlying algorithmic bottlenecks and search costs of a given task. When a proxy is carefully aligned with the objective computational demands of the problem, Complexity OoD remains a robust and indispensable diagnostic lens for distinguishing genuine algorithmic generalization from mere pattern matching and memorization.

## 3    The Duality of Learning and Reasoning under Complexity OoD

Traditional cognitive and AI paradigms often distinguish between learning (associated with System-1 rapid processing) and reasoning (associated with System-2 deliberative processing). However, treating these as mutually exclusive categories of *tasks* is misleading. The Complexity OoD framework reveals a fundamental duality: the distinction is not inherent to the task, but dictated by complexity. Specifically, tasks solvable by System-1 processing transform into challenges requiring System-2 processing when subjected to complexity-out-of-distribution pressures. Conversely, effective System-2 processing can be conceptualized as a sequence of learned System-1 decisions.

### 3.1    From Learning to Reasoning: When System-1 Fails, System-2 Emerges

Consider object recognition, a domain typically associated with rapid, perceptual processing. A model trained on images with a few, clearly separated objects learns to map input pixels to labels through what is effectively a high-dimensional pattern-matching function. Its generalization is evaluated on its ability to recognize new objects of the same classes. However, if we evaluate this model on instances with significantly higher representational complexity, for example, scenes with dozens of overlapping interacting objects, the fixed computational path of the model is no longer sufficient. The task is no longer one of simple recognition; it demands a process of segmentation, parsing of inter-object relations, and systematic composition of features, hallmarks of System-2 reasoning. This transition is equally evident along the computational axis. Consider the task of visual pathfinding or maze solving. At low complexity, such as a simple maze with few walls, determining connectivity between a start and end point can be a System-1 perceptual solvable task, often solved instantly via visual processing. However, as the maze scales in size and the solution path lengthens (increasing computational demand), the problem exceeds the capacity of perception. The agent is forced to switch to a serial, step-by-step visual tracing process. This effectively transforms what was a 'vision task' into a sequential 'search algorithm,' demonstrating how computational complexity alone can necessitate a transition from System-1 to System-2 dynamics. Thus, by pushing a System-1 processing solvable task into a Complexity OoD regime, we expose its hidden need for a reasoning-based approach.

### 3.2    From Reasoning to Learning: System-2 as Generalization over Solutions

Conversely, let us examine a System-2 processing demanding challenge, such as solving a multi-step mathematical problem. As established, a model confronting a computational Complexity OoD instance must generate a solution, with a greater computational depth than any it has observed during training. This perspective, however, naturally raises a critical question: If a model is designed with such a System-2 architecture, how does it "learn" from experience?

The answer lies in reframing the goal. A model that successfully generalizes is not just "thinking" in an abstract sense; it is demonstrating that it has learned a generative procedure for constructing valid solution. The effect of this learning becomes observable along two primary axes:

- **Improved Accuracy**: The most direct form of learning is an increase in the model's ability to generate the correct solution on its first attempt. Through training, its initial output becomes more likely to be valid, reflecting a better-calibrated internal model of the problem space.

- **Improved Efficiency**: A more profound form of learning emerges in settings where a verifier or oracle is available, allowing the model to test its proposed solutions. In such a scenario, learning is not just about being right immediately, but about reaching the correct answer more efficiently, with fewer attempts.

From this viewpoint, the solution produced by the model should be seen as a learned heuristic. The model's task is to navigate the vast search space of all possible solutions. A naive or untrained model might engage in a process akin to brute-force search, which is computationally intractable. A trained model, however, learns a heuristic function. This heuristic, itself a product of a System-1-like learning process, guides the construction of the solution by prioritizing more promising paths and pruning the search space.

- **Learning the Primitive Units**: An effective heuristic must operate on a well-defined set of building blocks. As discussed, these primitives must be sufficient to construct any solution and minimally redundant. Learning them involves an iterative process where units are first tuned on simple tasks and then co-adapted on more complex compositional problems. This provides the System-2 process with a powerful and expressive vocabulary.

- **Learning the Heuristic Function**: With a set of primitive units, the model must learn the heuristic function itself—the policy that dictates how to combine them. This is where the intuition of System-1 plays its most direct role. By being trained on successful and unsuccessful problem-solving traces, the model learns to recognize patterns that predict which sequence of units is most likely to lead to a correct solution. This learned function is what enables the model to bypass exhaustive search and efficiently generate solutions in practice.

In this light, the System-2 act of reasoning is powered by a deeply learned System-1-like intuition that guides its deliberate, step-by-step search. Achieving computational Complexity OoD is not an alternative to learning; it is the hallmark of a more profound and robust form of learning, one that masters the underlying structure of solutions, not just the surface statistics of problem-answer pairs. This mastery is what enables the model to generalize its solution-finding process to problems of arbitrary complexity.

### 3.3 Implications for Evaluation and Benchmarking

A critical question remains: what practical value do these dualities offer to the field, specifically through the lens of evaluation? We identify two primary implications that directly inform how future benchmarks and metrics should be designed:

- **Stress-Testing System-1 Processing via Complexity OoD:** The first duality suggests a methodology for stress-testing tasks that are conventionally solvable by System-1 processing. By subjecting these tasks to *Complexity OoD pressure*, we can perform a two-step analysis. First, we determine whether standard System-1 methods (e.g., direct feed-forward networks or simple perception modules) fail as complexity increases. Second, if they do fail, this framework guides us to identify what specific System-2 interventions (such as step-by-step reasoning, search, or decomposition) must be proposed to solve these high-complexity instances. This shifts the focus from merely "solving a task" to understanding the processing mode required at different complexity levels.

- **Diagnosing Reasoning via the Lens of Underlying Heuristics:** The second duality implies that existing System-2 methodologies should not be evaluated as black boxes. Instead, we must

examine them through the lens of the underlying heuristics acquired via System-1 learning. This perspective enables us to investigate whether fundamental limitations inherent to learning systems manifest within the reasoning process itself. For instance, we can assess whether the model's heuristic is compromised by data contamination, suffers from a lack of out-of-distribution (OoD) generalization, or is driven by spurious correlations rather than robust logical patterns.

## 4 Related Works

We note that from the outset, the field has repeatedly encountered scenarios that implicitly involve shifts in solution complexity, yet these were not framed explicitly as complexity out-of-distribution. In this section, we bring these threads under a single umbrella, our Complexity OoD perspective, and organize the review in three parts. First, we present work on representational and computational facets of complexity, tracing how variable-length, structured representations and variable-depth computation have been approached (e.g., object-centric and emergent language, adaptive computation, program-synthesize–style methods). Second, we review the recent trajectory in LLMs that integrates reasoning with learning via chain-of-thought prompting, test-time search and deliberation, repeated sampling with self-correction, reward-model supervision, and reinforcement learning. Third, we discuss the emerging trend of complexity-conditioned evaluation that probes long context, compositional structure, exploratory search, and long-horizon execution, advocating complexity-aware reporting instead of single aggregate scores.

### 4.1 Variable-length representation.

**Object-Centric Representation Learning:** Neural networks, particularly those based on conventional convolutional architectures, have demonstrated remarkable success in standard image recognition tasks. Nevertheless, they face pronounced limitations when applied to complex visual scenes comprising multiple objects and intricate inter-object relationships (Brady et al., 2023). As the complexity of a scene increases, these models often falter due to their inherently fixed-length representational capacity, a limitation known as the superposition catastrophe (Von Der Malsburg, 1986; Greff et al., 2020). This phenomenon refers to the network's inability to disentangle and separately encode multiple entities, resulting in entangled and ambiguous internal representations.

To address these shortcomings, recent research has increasingly focused on object-centric and structured representation approaches, with the Slot Attention mechanism emerging as a prominent example Locatello et al. (2020). Slot Attention combines low-level perceptual features from convolutional encoders with a fixed set of dynamic "slots" that compete via attention to bind to individual scene elements. Critically, this mechanism supports a flexible number of slots at inference time, allowing it to scale naturally with scene complexity and mitigating the superposition problem by enabling disentangled representations of discrete entities.

Despite these advancements, object-centric representation learning still faces fundamental challenges, particularly in acquiring causal and compositional representations with minimal supervision Didolkar et al. (2024); Mansouri et al. (2024); Kori et al. (2024); Kapl et al. (2025); Le Khac et al. (2024). Overcoming these obstacles is essential for the development of more cognitively grounded and System-2-compatible models, highlighting a fertile avenue for future research in artificial intelligence.

**Emergent Languages:** Language, a distinctive hallmark of human cognition, enables intricate communication, complex internal reasoning, and abstract thought. Inspired by these capabilities, researchers have developed the *Emergent Language* paradigm within artificial intelligence Havrylov & Titov (2017); Lazaridou et al. (2022; 2018); Peters et al. (2025). This field focuses on scenarios wherein multiple artificial agents participate in interactive, game-like tasks that encourage the spontaneous development of structured communication systems. Following the emergence of these novel linguistic forms, researchers analyze their compositional and syntactic properties to assess their functional and cognitive validity Lowe et al. (2019); Chaabouni et al. (2020); Carmeli et al. (2024).

Notably, emergent languages frequently exhibit discrete symbolic units (words) and variable-length message structures, enabling agents to convey information flexibly based on the complexity of their communicative

context Ueda & Washio (2021); Lee et al. (2024). This dynamic, context-sensitive flexibility directly aligns with solutions required for System-2 task such as generating detailed, variable-length descriptions based on situational complexity. Furthermore, the discrete, compositional nature of emergent languages closely mirrors the process of generating sophisticated solutions from basic, learnable semantic elements, reinforcing the connection to our conceptualization of processing needed for System-2 problems.

## 4.2 Variable-length computation

**Adaptive Computation Time:** One of the fundamental differences between humans and machine learning models is that the human response time to a problem can be a function of the difficulty of that problem, whereas, in machine learning models, the response time solely depends on the model architecture or the size of the input. For example, the longer the input sequence to a recurrent neural network (RNN), the longer it takes for the network to produce the final output. In other words, the human mind can devote more focus and attention to solving a problem with a more challenging input, something that traditional machine learning models are not capable of. To tackle this issue, Adaptive Computation Time (ACT), a mechanism embedding a halting unit within the RNN architecture, was introduced Graves (2016); Chowdhury et al. (2024). This unit dynamically decides the number of computational steps for each time step by outputting a halting probability, allowing the RNN to either continue processing or move to the next step. This enhancement led to improved performance in tasks like binary vector parity, integer addition, and real number sorting. The concept of a halting mechanism was extended to the transformer architecture, resulting in the Universal Transformer, which improved performance and accuracy on various algorithmic and language understanding tasks Dehghani et al. (2019a); Tan et al. (2024).

**Learning to Program:** Symbolic regression is a problem in machine learning that aims to discover the underlying mathematical expressions or symbolic equations that describe a given dataset. Unlike traditional regression methods that rely on predefined functional forms (based on neural network architecture), symbolic regression attempts to find the symbolic expressions directly from the data. Symbolic regression has a close relationship with variable-length computation. This relationship arises from the fact that the mathematical expressions discovered by symbolic regression can have varying lengths and complexities, depending on the nature of the underlying relationship in the data Biggio et al. (2021); Kamienny et al. (2022). This core idea was later more prominently implemented in the DreamCoder paper Ellis et al. (2021) . Notably, in DreamCoder, subprograms that frequently co-occurred could be combined and refactored, simplifying the search process across different programs. Recently, during the 2024 Arc Challenge, a significant number of top-ranked solutions used the Program Generation approach Chollet et al. (2024); Li et al. (2024b); Bonnet & Macfarlane (2024); Ouellette (2024); Singhal & Shroff (2024).

## 4.3 Some Shines of Integrating System-1 and System-2 via LLMs

While reasoning problems have traditionally been addressed outside the scope of learning-based methods, recent progress, driven especially by LLMs, has increasingly bridged the gap between the fields of learning and reasoning. More specifically, the generative nature of LLMs enables them to produce variable-length outputs, making them well-suited for tackling reasoning tasks that require flexible, structured solutions. In this section, we introduce recent efforts to solve reasoning problems by leveraging LLMs as a foundational infrastructure.

**Chain of Thought (CoT):** For reasoning tasks, LLMs can be asked to write the solution step-by-step before providing the final answer Wei et al. (2022b); Xia et al. (2025). This can enable the language model to generate longer solutions for more complex problems by generating tokens sequentially. The CoT idea helped significantly improve the performance of language models on some reasoning tasks. However, since LLMs are still confined to left-to-right decision-making processes (without backtracking) during inference, they can fall short in System-2 tasks that require exploration, strategic lookahead, or where initial decisions play a pivotal role He et al. (2025). This means that for certain reasoning tasks, LLMs still faced challenges.

**LLMs and Search:** Since the trained LLMs by a System-1 approach can not guarantee to solve all reasoning problems naively by the CoT approach as discussed above, some approaches that need to explore during the test time in order to find the output have been introduced. Ideas such as Tree of Thought (ToT) and Graph of Thought (GoT) allow LLM to branch and generate the solution step-by-step through a search process during the inference time Yao et al. (2024); Besta et al. (2023); Koh et al. (2024); Zhang et al. (2024); Chen et al. (2024); Bi et al. (2024); Yu et al. (2024); Wang et al. (2025); Ding et al. (2025). ToT allows LLMs to perform deliberate decision-making by considering multiple different reasoning paths and self-evaluating choices to decide the next course of action, as well as looking ahead or backtracking when necessary to make global choices. In case of failure, it has the ability to backtrack and construct a new solution. This concept is clearly analogous to the concept of learn-to-search.

**LLMs and repeated sampling:** LLMs, as probabilistic generative models, offer the capability to generate a diverse range of step-by-step solutions. LLMs achieve this through *repeated sampling*, a technique that increases the likelihood of generating an optimal response Li et al. (2022); Rozière et al. (2023). Common sampling strategies in LLM inference include top-p (Nucleus Sampling) and top-k sampling, which enable the parallel generation of multiple candidate outputs. By leveraging repeated sampling, LLMs enhance their chances of producing accurate and high-quality responses Brown et al. (2024), akin to how algorithm designers iteratively refine their solutions to improve computational efficiency. Self-correction is a test-time computation method that allows LLMs to iteratively revise and refine generated results using external or internal feedback Shinn et al. (2023); Ye & Ng (2024); Madaan et al. (2023). A critical aspect of this iterative process is the implementation of evaluation and verification strategies, which ensure the effectiveness of repeated sampling and contribute to the overall reliability of the generated outputs. Selecting the most frequent answer as a verification strategy can enhance accuracy, particularly in approaches like self-consistency CoT Wang et al. (2022); Li et al. (2024a); Lin et al. (2023). Moreover, the reward models presented below offer a systematic approach to assessing generated reasoning traces.

**LLMs and Reward Models:** Reward models are primarily categorized into two types: Outcome-based Reward Models (ORMs) and Process-based Reward Models (PRMs). ORMs evaluate solutions based solely on the correctness of the final Chain-of-Thought (CoT) output, and thus provide a relatively coarse feedback signal Cobbe et al. (2021); Bai et al. (2022). In contrast, PRMs are trained on finer-grained annotations that assess the validity of each intermediate reasoning step, enabling them to localize errors and provide richer supervisory signals Uesato et al. (2022); Lightman et al. (2023); Wang et al. (2023b). Recent studies show that PRMs significantly outperform ORMs in domains such as mathematics and code generation, as their localized feedback improves both reliability and robustness Lightman et al. (2023); Zhou et al. (2024). However, PRMs are costly to construct since they require high-quality, step-level annotations, often from domain experts, making scalability a central challenge Wang et al. (2023b); Shi et al. (2024). To alleviate this, automated annotation techniques have been proposed, including Monte Carlo Tree Search (MCTS)-based labeling Wang et al. (2024), synthetic reasoning traces Li et al. (2023), and weak-to-strong generalization frameworks where smaller, trusted models generate labels for training larger models Burns et al. (2023). Importantly, PRMs can also serve as heuristic functions to guide search over candidate reasoning trajectories, closely resembling neural-guided search in program synthesis and theorem proving Polu & Han (2022); Chen et al. (2023). Beyond training, both ORMs and PRMs are increasingly employed at inference time to discriminate between desirable and undesirable outputs, for instance through reranking or rejection sampling across multiple LLM candidates Uesato et al. (2022); Wang et al. (2023b); Zhou et al. (2024). This dual utility—providing step-level supervision and enabling inference-time selection—highlights reward models as a key component in advancing the reliability and generalization of reasoning-capable LLMs. On the other hand, reward models can be employed in a Reinforcement Learning (RL) pipeline too as discussed below.

**LLMs and RL** : A recent approach proposed in several studies Wang et al. (2024); Setlur et al. (2024); Zelikman et al. (2022); Huang et al. (2023) is fine-tuning LLMs by an RL paradigm using the CoTs generated by the LLMs themselves and evaluated by verifiers or reward models (mentioned above) which provide supervision feedback. Unlike Supervised Fine-Tuning (SFT), which tends to overfit to training data and struggles with generalization to out-of-distribution scenarios Chu et al. (2025); Singhal et al. (2023), RL methods gen-

eralize to unseen situations more effectively by optimizing policies against adaptive reward signals. While the community has, to date, often preferred process reward model (PRM)-based verifier methods (especially after the success of the o1 model), several new directions have emerged. For example, DeepSeek R1 Guo et al. (2025) and related work on verifier-free RL Zhou et al. (2025a) demonstrate that large-scale models can be trained via pure RL with only simple correctness and structural rewards, eliminating explicit verifiers while maintaining competitive performance. Other recent studies propose more efficient reinforcement learning variants, such as contrastive CoT-based reinforced fine-tuning (CARFT) Liu et al. (2025), or reinforcement-learning–based knowledge distillation that leverages multi-branch reasoning structures (RLKD) Sun et al. (2025a). Despite their simplicity, these approaches rival verifier-based pipelines like o1 Guo et al. (2025); Wang et al. (2024); Bai et al. (2022), highlighting the versatility of RL for reasoning. From a Complexity OoD perspective, RL approaches are especially significant since they inherently encourage the model to allocate computational resources dynamically in proportion to the complexity of the problem encountered at inference. This enables the emergence of an "aha moment," where the model recognizes when its initial reasoning path is insufficient for a complex scenario and accordingly invests greater computational effort, revises its reasoning steps, or backtracks to construct a more suitable solution. Zan et al. (2025).

### 4.4 Considering Task Complexity in the Evaluation of LLMs

After initial successes on short, well-structured problems, LLMs have very recently been applied to substantially more complex tasks in reasoning, planning, and software engineering, whose defining characteristics include long context, step-by-step decision making, and strategic planning. Foundational evidence has already cautioned that Transformers, the backbone of most LLMs, struggle as compositional and structural complexity increases, placing limits on scale-alone solutions to systematic generalization Dziri et al. (2023). Complementing this, Zhou et al. (2025b) systematically scales both context length and reasoning difficulty and observes reliability drop-offs under increasing length and complexity, motivating compute-aware protocols and complexity-conditioned reporting. Complexity-binned analyses in Shojaee et al. (2025) further show that models often collapse at higher problem complexity, a pattern consistent with contamination-inflated performance on easier instances and underscoring the need to evaluate along the complexity axis rather than with a single average. Sinha et al. (2025) disentangles planning from execution and demonstrates that even when the correct algorithmic plan is provided, models frequently fail over long execution horizons due to brittle state tracking and procedural fidelity, calling for metrics that couple horizon length with step-level correctness. In parallel, there is a clear trend toward benchmarks that explicitly probe these complex settings, including Sun et al. (2025c) for exploratory, compositional, and transformative mathematical generalization, Sun et al. (2025b) for procedural correctness via simple program execution, and Qiu et al. (2025) for long-context, cross-file software engineering with multi-stage decision making, collectively reinforcing that evaluation of LLMs must be conditioned on task complexity and long-horizon execution demands.

## 5 Constructing the Foundations of Complexity OoD Generalization

In the preceding sections, we have introduced Complexity OoD as a novel conceptual framework for understanding and evaluating reasoning. We demonstrated how this lens unifies the seemingly disparate concepts of learning (System-1) and reasoning (System-2), revealing a fundamental duality between them. Furthermore, by surveying various research directions, we have shown how the field is already implicitly grappling with different facets of the Complexity OoD challenge. Beyond this theoretical formulation, however, we must address the practical implications. Having accepted the importance of the Complexity OoD challenge, what changes must we make to our research trajectories, model development practices, and evaluation methodologies? This section proposes several concrete, actionable shifts in perspective and priority that can guide the field toward building more robust and generalizable reasoning agents.

### 5.1 Rethinking Evaluation: Tasks and Benchmarks

System-1 neural network architectures have existed for years, but the rapid evolution of deep learning began with the introduction of the ImageNet dataset in 2012 Russakovsky et al. (2014) . The event referred to as the ImageNet moment made the ImageNet dataset gain significant attention as the first large-scale dataset

for benchmarking deep-learning vision networks. We believe that to ignite the progress of System-2, there must be a spark in creating tasks and benchmarks specifically tailored for it. In other words, System-2 needs its own version of the ImageNet moment. One example of such a benchmark is the ARC (Abstraction and Reasoning Corpus) Challenge proposed by François Chollet, which consists of tasks designed to evaluate more advanced reasoning capabilities beyond pattern recognition Chollet (2019). Of course, defining a foundational task with maximum inclusivity for System-2 is a non-trivial and complex matter, requiring extensive investigation. Nevertheless, alongside this main path, a parallel path can be pursued where tasks and benchmarks of System-1 are addressed using an approach inspired by System-2. For example, consider image classifiers that, upon receiving an image, attempt to generate the output over a variable number of steps depending on the complexity of the image.

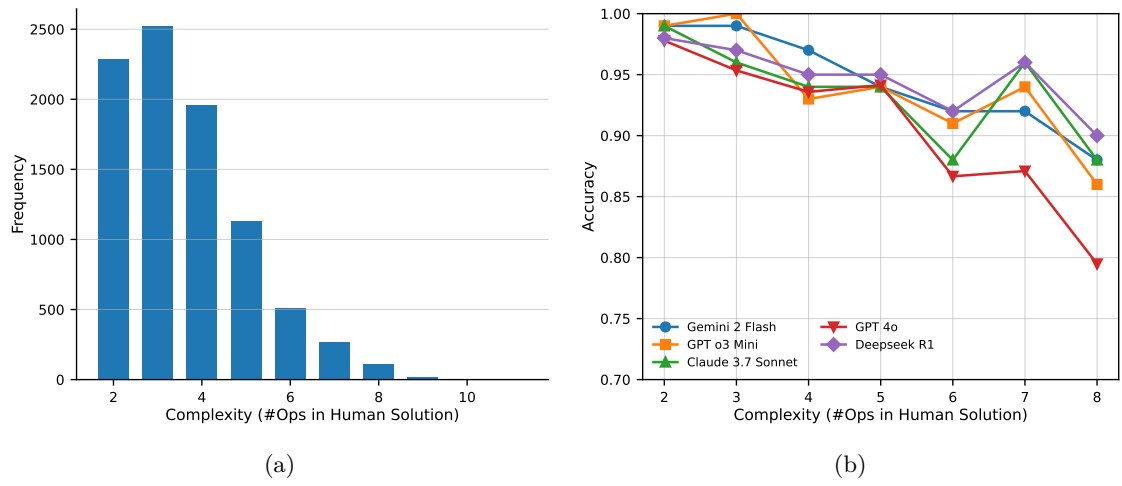

(a)                                                          (b)

Figure 3: (a) The frequency distribution of problem complexity in the GSM8K dataset, measured by the number of arithmetic steps in reference solutions. Most problems are simple, leading to a strong imbalance in the dataset. (b) Model accuracy on GSM8K across different complexity levels, illustrating that as problem complexity increases, model performance drops (especially for non-reasoning models) highlighting the key challenge of complexity out-of-distribution (OoD) generalization. This analysis reveals limitations obscured by average-case metrics and motivates the need for complexity-aware evaluation in benchmarking reasoning ability.

Consider the GSM8K dataset, a widely used benchmark consisting of elementary-level math word problems designed to test arithmetic reasoning. While models often achieve high average accuracy on this dataset, it is generally treated as a "solved" benchmark. However, when we analyze models' accuracy based on the inherent complexity of the samples, a different picture emerges. We define the complexity of each sample as the number of arithmetic operations in its corresponding human-written solution. As shown in Figure 3a, the complexity distribution is highly imbalanced—approximately exponential—where simpler problems are much more frequent than complex ones. Consequently, the standard evaluation metric (average accuracy) is heavily biased toward these simpler cases, potentially providing an overly optimistic view of model capabilities. As shown in Figure 3b, when we break down performance by complexity levels, a consistent trend appears: as complexity increases, accuracy decreases. Even though all problems are elementary in nature, LLMs exhibit drops in accuracy for higher-complexity samples. This effect reveals that performance metrics which aggregate over all examples obscure the true generalization capabilities of the model. Furthermore, the rate at which accuracy deteriorates varies across models. For instance, reasoning-oriented models such as DeepSeek-R1 and GPT-o3-mini show a gentler degradation curve, while non-reasoning models like GPT-4o, break more sharply as complexity rises. This widening performance gap at higher complexity levels reveals an important insight: evaluating models by their failure rate under increasing complexity provides a clearer and more nuanced view of complexity generalization. Thus, incorporating complexity-aware evaluation into benchmarks like GSM8K highlights the importance of Complexity OoD. It enables us to distinguish between

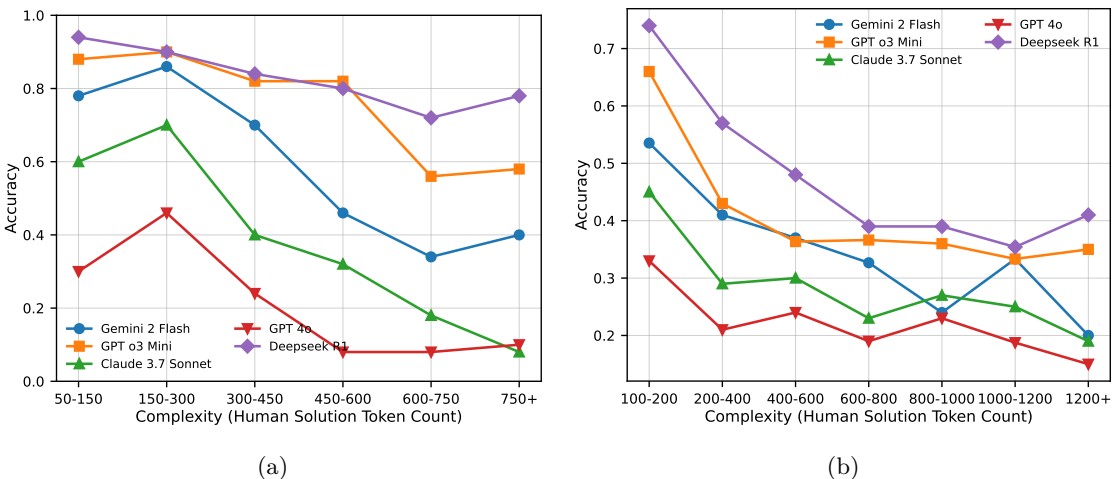

(a)                (b)

Figure 4: (a) Accuracy of Language Models on AIME by Human Solution Token Length. Complexity is estimated by the number of tokens in provided human-written solutions for each problem, used here as a proxy for the length and intricacy of multi-step reasoning required. Accuracy declines notably with increasing solution length for all models; however, models designed for advanced reasoning (DeepSeek-R1, GPT-o3-mini) maintain higher accuracy and exhibit a gentler degradation compared to more general-purpose models. (b) Accuracy of Language Models on Omni-MATH by Human Solution Token Length. Here too, as the solution complexity (token count) rises, model accuracy drops, especially for models not explicitly optimized for complex reasoning. The pattern reinforces the importance of evaluating models on complexity out-of-distribution (OoD) instances.

models that merely memorize common patterns and those that demonstrate robust, systematic generalization under increasing reasoning demands.

Similar complexity-aware analysis can be extended to other reasoning benchmarks, such as AIME and Omni-MATH. The AIME dataset consists of problems from the American Invitational Mathematics Examination, featuring high-school level olympiad-style questions that require multi-step reasoning and are more challenging than those in GSM8K. The Omni-MATH dataset goes further, aggregating a wide range of advanced mathematical problems from various national and international mathematics olympiads. These questions are often regarded as some of the most difficult reasoning problems available for benchmarking. Unlike GSM8K, where complexity can be clearly defined by counting the number of arithmetic operations in a human-written solution, measuring complexity in datasets like AIME and Omni-MATH is more challenging. These datasets often lack standardized, fine-grained, step-by-step human annotations. To approximate reasoning complexity in these cases, we use the number of tokens in the human-written solution as a proxy, interpreting longer solutions as indicative of more elaborate reasoning procedures. After computing the token lengths for each solution, we group the test samples into buckets of fixed size (e.g., 150 or 200 samples per bin) to equalize comparison and compute accuracy for each complexity bin. As shown in Figure 4, the pattern observed in GSM8K becomes even more pronounced: model accuracy degrades more steeply as problem complexity increases. Importantly, reasoning-oriented models such as DeepSeek-R1 and GPT-o3-mini consistently outperform others across all complexity levels and exhibit a more gradual decline in performance. This aligns with our broader finding that RL-trained models generalize better to complexity OoD cases, supporting the hypothesis that reinforcement learning enhances a model's ability to dynamically allocate computation and adapt to harder reasoning tasks. These observations underscore that evaluating model robustness across complexity bins is essential for understanding generalization. Metrics that ignore this aspect fail to reflect true System-2 capabilities. In this light, benchmarks like AIME and Omni-MATH offer critical testbeds for studying models' behavior in the presence of symbolic complexity, pushing us closer toward evaluating—and achieving—System-2-level generalization.

A natural and important objection to the complexity-binned analysis above is that observed accuracy degradation might be driven not by the depth of the required computation, but by a surface-level confounder: harder problems often have longer problem statements, more unusual phrasing, or less common numerical values. If a model's failures tracked *question* length rather than *solution* complexity, then our analysis would merely re-describe length generalization rather than diagnose Complexity OoD. To disentangle these factors, we directly test whether accuracy degradation is more strongly associated with the length of the required solution (our proxy for computational depth) or with the length of the input question (the surface confounder).

We analyze five state-of-the-art models—Claude 3.7 Sonnet, DeepSeek-R1, Gemini 2 Flash, GPT-4o, and GPT-o3-mini—across three mathematical reasoning datasets spanning a wide difficulty range (GSM8K, AIME, and Omni-MATH). For each problem, we measure (i) the human-written *solution* token count, used as a proxy for computational complexity, and (ii) the *question* token count, the surface-level confounder. We then compute Pearson correlations between these quantities and per-instance correctness. Crucially, the correlation between question and solution length, $\mathrm{Corr(Sol, Quest)}$, lets us gauge how entangled the two axes are in each dataset.

Table 1: Correlation analysis isolating solution complexity from question-length confounders. Across all datasets and models, accuracy degradation correlates *more strongly* with solution length (computational depth) than with question length (surface confounder). The strongest (most negative) accuracy correlation in each row is shown in **bold**. The effect is cleanest on Omni-MATH, where the two length axes are nearly decoupled ($\mathrm{Corr(Sol, Quest)} \approx 0.08$), yet solution length remains predictive of failure while question length does not.

| Dataset | Model | Corr(Sol, Quest) | Corr(Sol, Acc) | Corr(Quest, Acc) |
|---------|-------|:----------------:|:--------------:|:----------------:|
| AIME | Claude 3.7 Sonnet | 0.311 | **−0.390** | −0.187 |
| | DeepSeek-R1 | 0.311 | **−0.162** | −0.086 |
| | Gemini 2 Flash | 0.311 | **−0.332** | −0.260 |
| | GPT-4o | 0.311 | **−0.201** | −0.092 |
| | GPT-o3-mini | 0.311 | **−0.279** | −0.130 |
| Omni-MATH | Claude 3.7 Sonnet | 0.082 | **−0.126** | −0.033 |
| | DeepSeek-R1 | 0.082 | **−0.156** | −0.072 |
| | Gemini 2 Flash | 0.081 | **−0.191** | −0.072 |
| | GPT-4o | 0.082 | **−0.103** | −0.008 |
| | GPT-o3-mini | 0.082 | **−0.150** | −0.115 |
| GSM8K | Claude 3.7 Sonnet | 0.549 | **−0.072** | −0.051 |
| | DeepSeek-R1 | 0.549 | **−0.073** | −0.035 |
| | Gemini 2 Flash | 0.549 | −0.133 | **−0.137** |
| | GPT-4o | 0.549 | **−0.267** | −0.161 |
| | GPT-o3-mini | 0.549 | −0.119 | **−0.139** |

The results in Table 1 consistently support the Complexity OoD interpretation. Across thirteen of fifteen model–dataset combinations, the magnitude of the (negative) correlation between accuracy and *solution* length exceeds that between accuracy and *question* length. In other words, the depth of the computation a problem demands is a better predictor of failure than how long the problem reads. This pattern is most striking on Omni-MATH, where question and solution length are almost decoupled ($\mathrm{Corr(Sol, Quest)} \approx 0.08$): here question length is essentially uninformative about correctness (e.g., −0.008 for GPT-4o), whereas solution length retains a clear negative relationship with accuracy (−0.103 to −0.191). On AIME, the effect of solution complexity is frequently about double that of question length (e.g., −0.390 vs. −0.187 for Claude 3.7 Sonnet). Even on GSM8K—where the two length axes are highly entangled ($\mathrm{Corr(Sol, Quest)} = 0.549$), making the cleanest separation impossible—solution length is at least as predictive of failure as question length for every model.

These findings reinforce that the degradation of System-2 performance is governed primarily by the depth of the latent computation required to reach the answer, not by the surface form of the input. This both

validates solution-length as a usable (if imperfect) computational-complexity proxy in these domains and, more broadly, provides a concrete instance of the proxy-validation procedure advocated in Section 2.4: measure $\text{Corr}(\text{proxy}, \text{surface length})$ and confirm that the proxy predicts failure *beyond* what surface length explains.

In light of the above, reporting a single aggregate score (e.g., average accuracy) without conditioning on sample complexity is insufficient for assessing reasoning ability. Evaluation should include *complexity-aware* measures that condition performance on a binned complexity proxy $c$ (e.g., operation count, hop/proof depth, solution token length). The central diagnostic is the Complexity–Performance Curve (CPC), which plots accuracy $A(c)$ versus $c$ with 95% confidence intervals; a bin-uniform area under this curve (AUC–CPC) summarizes performance across complexity independently of dataset skew. Two additional summaries make the CPC actionable: (i) the *breakdown point* $c^\star = \min\{ c : A(c) < \alpha \}$ for a target threshold $\alpha$ (e.g., 0.5 or chance$+\varepsilon$), identifying the smallest complexity at which the model collapses; and (ii) the *degradation slope*, obtained by fitting $A(c) \approx \beta_0 + \beta_1 c$ and reporting $-\hat{\beta}_1$ as the rate of decline with complexity. Together, these complexity-aware metrics expose failure modes that average accuracy obscures and provide a principled basis for evaluating Complexity OoD generalization.

### 5.2 Rethinking Training: Supervision Paradigms

The challenge of training a System-2 model is fundamentally the challenge of supervising the synthesis of its solution paths. Drawing parallels with System-1, which encompasses supervised, unsupervised, and self-supervised regimes, we can categorize the supervisory landscape for reasoning based on the nature and granularity of the available feedback. This distinction is crucial, as the chosen paradigm dictates the scalability of learning and the types of reasoning skills a model can acquire.

**Strong Supervision: Learning from Exemplary Solution Traces.** This is the most direct form of supervision, where the training data consists of triplets: `(problem, correct solution path, final answer)`. This approach is akin to a student being shown an explicit, step-by-step worked example. It offers a powerful and precise learning signal, making it highly effective for training the solution generation component. However, its primary limitation is the scarcity and high cost of data. Creating high-quality, step-by-step reasoning traces requires significant human expertise and effort, making this paradigm powerful in theory but difficult to scale in practice.

**Weak Supervision: Learning from Final Outcomes.** A far more common and scalable scenario is one where supervision is only available for the final answer, with training data consisting of pairs: `(problem, final answer)`. In this paradigm, the intermediate reasoning trace is a latent variable that the model must infer. This transforms the learning problem into a difficult credit assignment challenge: if the final answer is wrong, which of the intermediate steps was flawed? This setting is a natural fit for Reinforcement Learning (RL), where the correctness of the final answer serves as a sparse reward signal to guide the exploration of the vast search space of possible solution paths.

**Meta-Learning: Learning to Discover Reusable Cognitive Primitives.** Instead of learning to solve tasks in isolation, we can aim higher: learning to learn how to reason across a diverse range of problems. In this meta-learning paradigm, the model is exposed to a multitude of different tasks. The goal is not just to master each task, but to force the model to discover the shared, underlying atomic units and compositional rules that are useful across all of them. By learning to induce these reusable cognitive primitives, the model acquires a foundational, extensible toolkit for reasoning, enabling it to tackle novel problems more effectively. This aligns with how humans build up a library of problem-solving techniques.

**Self-Supervised Learning: Creating Supervision from Unlabeled Data.** Drawing inspiration from the success of self-supervision in System-1, we can devise analogous objectives for learning in the domain of solutions. Given a large corpus of unlabeled solutions or reasoning traces (e.g., from open-source code repositories or scientific papers), we can train the heuristic function for the solution generator. For example, a "masked solution modeling" objective could involve masking a sub-routine or a logical step within a solution

and training the model to predict the missing part from the surrounding context. A contrastive objective could train the model to recognize that two different-looking solutions are semantically equivalent (e.g., they implement the same algorithm) or that a slight change to a solution drastically alters its function. Such methods could allow the System-2 machinery to learn the structure and semantics of valid reasoning without requiring paired problem-solution data.

### 5.3   Rethinking Methods: Inductive Biases for Complexity Out-of-Distribution

The remarkable success of deep learning architectures stems from their powerful inductive biases, which align with the inherent structure of data (e.g., translation equivariance in CNNs, sequentiality in RNNs). Just as generalization over any form of out-of-distribution data requires appropriate inductive biases, overcoming the Complexity OoD challenge is fundamentally dependent on them. This necessity is especially critical for Complexity OoD because, as we established earlier, it cannot be resolved merely by scaling training data. For any training set, regardless of the complexity of its instances, one can always construct a test set with problems whose solution complexity exceeds that of the training distribution.

Therefore, the core of any effective inductive bias for Complexity OoD must be to enable unbounded capacity at inference time, for both representation and computation. This means moving beyond architectures with fixed computational graphs and static representational limits. Current models, including the Transformer, possess general-purpose sequence processing capabilities but lack the specific priors needed to efficiently learn and generalize in the structured, combinatorial space of solutions.

To theoretically formalize why specific inductive biases are required rather than merely beneficial, consider the operational limits of standard architectures. A vanilla Transformer possesses a fixed computational depth bounded by its layer count and constrained context window. If we define its maximum sequential reasoning capacity as $N$ operations, the model is mathematically bottlenecked: if a Complexity OoD test instance requires a minimal sequential reasoning depth of $N+1$ steps, the standard model fundamentally cannot execute it, regardless of the volume of training data or the effectiveness of the optimization process. Incorporating mechanisms like adaptive computation, test-time search, or external memory does not absolutely guarantee that the model will *learn* the correct target algorithm, as optimization failures can still occur. However, they act as **necessary structural prerequisites**. By allowing a model to dynamically execute computational steps or expand its state-tracking, these biases fundamentally decouple computational depth from parameter count. This endows the model with the theoretical expressivity required to represent and execute solution paths of arbitrary algorithmic complexity $(N + k)$ that strictly exceed the training distribution.

We identify three crucial areas where new inductive biases are essential to unlock this dynamic, unbounded capacity:

**Unbounded Representational Capacity and Solution Structure:**   System-1 models operate on fixed-dimensional vectors. In contrast, System-2 reasoning requires representing solutions of variable and potentially unbounded length and complexity. This necessitates a shift from feature spaces to solution spaces. A powerful inductive bias is one that favors modular and compositional representations of solutions. Instead of treating a solution as a flat sequence of tokens, architectures should be biased towards representing them as structured objects like Abstract Syntax Trees (ASTs) or computational graphs. This structural bias would allow the model to learn and reuse sub-routines (functions or modules), a cornerstone of efficient solution synthesis and a key mechanism for generalizing to more complex problems by composing known building blocks in novel ways Ellis et al. (2021).

**Adaptive Computational Depth:**   A defining feature of human reasoning is the ability to allocate more computational effort to harder problems. Most neural networks, however, have a computational depth fixed by their architecture. To overcome computational Complexity OoD, models must possess an inductive bias for adaptive computation. Mechanisms like the Halting Unit in Adaptive Computation Time (ACT) Graves (2016) or the recurrent nature of Universal Transformers Dehghani et al. (2019a) are early examples. Future research should explore more potent biases for learning recursive and iterative procedures. An architecture with a native bias for recursion could learn the general algorithm for a task (e.g., factorial or graph traversal)

from a few examples, enabling it to execute the algorithm for any required depth at inference time, far beyond what was seen during training.

**External Memory, Statefulness, and Execution Fidelity:** Complex, multi-step reasoning often requires not only constructing a correct algorithm but also faithfully executing it by meticulously tracking intermediate results and state changes. The human brain relies on working memory for this. Recent studies provide compelling evidence that a core failure mode of LLMs in reasoning tasks is not just an inability to devise a correct algorithm, but an inability to execute one Shojaee et al. (2025); Sinha et al. (2025). These works demonstrate that even when provided with an explicit, correct algorithm, LLMs often fail by "forgetting" the current state of the problem and proposing invalid actions.[3]

This highlights a critical bottleneck, the transient activations of a Transformer are insufficient to serve as a reliable working memory for complex, stateful procedures. Therefore, an inductive bias for interacting with an external memory structure is not just beneficial, but essential. Architectures like the Neural Turing Machine Graves et al. (2014) provided an early proof-of-concept. By incorporating a bias for reading from and writing to a persistent, stateful memory, a model is no longer required to encode the entire history of its computation within its internal activations. This allows it to offload intermediate products of its reasoning process, freeing up internal resources. More importantly, it enables execution fidelity. Augmenting LLMs with external tools, such as a dedicated memory to store state variables or verifiers that check the validity of each action before execution, can directly address this failure mode. Such a mechanism would empower models to construct and faithfully execute longer and more intricate solution, a prerequisite for overcoming computational Complexity OoD.

### 5.4 Rethinking Problems: Revisiting Learning Challenges in the System-2 Domain

The shift to a System-2 paradigm does not erase the fundamental challenges of machine learning; rather, it recasts them in a new, more abstract domain. As we have argued, System-2 is not a faculty divorced from System-1. Instead, the process of reasoning fundamentally relies on learning: a System-1-like mechanism is learned to act as a heuristic, guiding the search for and construction of a valid solution path.

This deep entanglement means that the System-2 reasoning process inevitably inherits the well-documented "pests" and pathologies of its underlying learning component. Consequently, achieving reliable and trustworthy reasoning is contingent upon our ability to identify, redefine, and address these foundational learning challenges as they manifest in the solution synthesis domain. A robust System-2 agent must be robust not just in its final output, but throughout its entire generation process. In what follows, we illustrate how several canonical learning challenges re-emerge in this new context:

**Spurious Correlation and Shortcut Learning in Solution Synthesis:** Shortcut learning in System-2 can be more insidious than in System-1. A model might learn a spurious correlation not between input pixels and a class label, but between superficial textual cues in a problem statement and the structure of the solution. For instance, it might learn that the word "more" always implies an addition operation, failing on problems where "more" is used in a comparative but non-additive context. This is not a failure of calculation, but a failure of learning the correct causal mapping from problem semantics to solution logic.

**Semantic Adversarial Robustness:** Adversarial attacks in System-1 involve small, human-imperceptible perturbations to inputs (e.g., pixels). The equivalent in System-2 is a semantic perturbation: a small, meaning-preserving change to the problem statement that causes a catastrophic failure in the generated solution. For example, changing "Alice has 5 apples, Bob has 3" to "Bob has 3 apples, and Alice has 5" should yield the same solution for a query about the total. A brittle model might be highly sensitive to such word-order variations. Future benchmarks must explicitly test for this semantic robustness, evaluating whether models are learning abstract algorithms or just fragile patterns of text-to-solution mapping.

---

[3]For instance, when tasked with solving the Tower of Hanoi puzzle, models may correctly follow the rules for several steps but then suggest a move that is physically impossible given the current configuration of disks, indicating a loss of state representation Shojaee et al. (2025).

Beyond meaning-preserving edits, a distinct attack surface targets the model's internal assessment of complexity: inputs can be crafted to induce a misestimation of the required solution depth, either under-allocating computation on hard instances or expanding the reasoning trace unnecessarily on easy ones. Concretely, adversaries may insert semantically irrelevant distractors or nested quantifiers (e.g., extra entities and relations in VQA, gratuitous case splits or variable renamings in math and code) that inflate the apparent branching factor, yielding longer solution traces, higher runtime, and even compute-amplification or DoS-style failure modes without changing the underlying task. Future benchmarks should explicitly include complexity-controlled paraphrases and distractor-augmented variants, and training should encourage calibrated halting and penalize needless computation (e.g., via step-cost regularization, process-level verification, and adversarial training that targets the model's complexity estimator).

**Catastrophic Forgetting of Reasoning Skills:** In continual learning, catastrophic forgetting occurs when a model trained sequentially on new tasks forgets knowledge acquired earlier. In the System-2 setting, forgetting can be triggered not only by task identity but by shifts in domain and, crucially, in the distribution of required solution complexity. A model first trained on domain A with a characteristic complexity profile and then fine-tuned on domain B with a systematically different complexity profile, either higher or lower, might become unable to reason effectively on the original domain. The second stage can overwrite procedures calibrated to the first domain's complexity regime: training on simpler, template-like instances can erode multi-step planning previously learned, while training on longer, search-heavy instances can inflate halting thresholds and induce unnecessary overthinking on easy cases. This interference across domains and complexity levels blocks the accumulation of a broad reasoning repertoire. Addressing it requires complexity-aware continual learning of algorithmic skills—preserving and indexing skills by domain and solution depth, employing (synthetic) replay across complexity bins, stabilizing reasoning primitives and halting policies via regularization, and modularizing architectures, so that new capabilities are integrated without overwriting foundational ones.

**Poor Calibration and Uncertainty in Multi-Step Reasoning:** In System-1, calibration refers to how well a model's predicted confidence matches its actual correctness. A poorly calibrated model is "confidently wrong." In System-2, this problem becomes more nuanced and critical. A model might generate a multi-step solution, but does it know when it is "stuck" or when a specific reasoning step is likely incorrect? A well-calibrated reasoner should be able to express uncertainty not just about the final answer, but about the intermediate steps of its own reasoning trace. For example, it should be able to signal "I am uncertain about this logical deduction." This lack of self-awareness about its own reasoning process prevents the model from efficient backtracking, asking for help, or strategically allocating more search effort to weaker parts of its solution path.

By explicitly addressing these revisited challenges, we can ensure that our pursuit of System-2 reasoning does not simply replicate the brittleness of System-1 models at a higher level of abstraction, but leads to truly robust and generalizable intelligence.

## 6   Conclusion

In this work, we confronted a fundamental, yet often overlooked, challenge in modern artificial intelligence: the absence of a clear and robust framework for measuring genuine reasoning ability in AI models. We argued that existing evaluation paradigms, largely inherited from System-1 pattern recognition tasks, are insufficient for System-2 reasoning. They often rely on average-case performance metrics that are susceptible to data contamination and fail to provide a fine-grained understanding of a model's true capabilities, particularly its breaking points when faced with novel, complex problems.

To address this gap, we introduced Complexity Out-of-Distribution (Complexity OoD) as a new conceptual framework. We proposed a fundamental reinterpretation: that "reasoning ability" is best understood and measured as a model's capacity to generalize to problem instances whose minimal required solution complexity, be it in representation or computation, lies significantly outside the distribution of its training data. This perspective shifts the focus from simply verifying final answers to assessing the underlying generative process of problem-solving.

The primary power of the Complexity OoD framework is its unifying nature. First, it dissolves the rigid dichotomy between System 1 (learning) and System 2 (reasoning), revealing a profound duality: System-1 tasks become System-2 challenges under complexity pressure, and successful System-2 performance can be reconceptualized as a sophisticated form of learning to generalize over the structure of solution paths. Second, it provides a more robust and diagnostic lens for evaluation, allowing us to move beyond superficial accuracy scores and instead measure how gracefully a model's performance scales with problem complexity.

Finally, our framework illuminates several concrete and critical future research directions necessary for building the next generation of reasoning agents. These include:

- Rethinking Benchmarks: We must move towards complexity-aware evaluation, designing benchmarks that explicitly test for Complexity OoD and analyzing model performance across stratified levels of difficulty.

- Exploring New Supervision Paradigms: Just as System-1 learning evolved from supervised to self-supervised paradigms, we must explore new forms of supervision for System-2. This involves moving beyond simple outcome-based rewards to process-based supervision, and crucially, developing methods for learning to reason in unsupervised or minimally supervised settings where step-by-step guidance is unavailable.

- Inventing New Inductive Biases: Achieving Complexity OoD is not a matter of scale but of architecture. The development of novel inductive biases, such as those for adaptive computation, external memory, and abstraction, is paramount for creating models with the capacity for unbounded reasoning.

- Revisiting Foundational Challenges: Classic machine learning problems like spurious correlation, catastrophic forgetting, and adversarial robustness do not disappear; they re-emerge in the domain of solution synthesis and must be redefined and tackled to build truly reliable systems.

Achieving System 2-level artificial intelligence will not come from simply scaling up existing models. It demands a fundamental shift in how we evaluate, build, and train models, a shift that equips them with the right inductive biases for generalizing across complexity. By adopting this new lens, we can move beyond measuring performance on static benchmarks and begin to cultivate robust, genuine reasoning. This is the path that will lead to a new generation of AI that does not just learn, but truly thinks.

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
