# OpenReview forum: "Bridging Reasoning to Learning: Unmasking Illusions using Complexity Out-of-Distribution Generalization"
_TMLR — Accepted by TMLR_

### Review · Reviewer_DDRX · 2025-11-30

**Summary Of Contributions:**

## Summary of contributions
The core contribution of this paper is the idea of generalization to problems with higher complexity demands along the axes of reasoning and representation. The title is a bit misleading -- “complexity OOD” paints the picture of a probability distribution along a 1-D complexity axis that is different across training and inference time; however, the text of this paper focuses on generalization from lower to higher complexity, so a better fitting name would be “generalization to higher complexity distribution”.

## Strengths
1. Most claims are well substantiated with convincing evidence and sufficient groundwork. The clear examples in Fig 1 (roman numerals, vqa), Section 2.1 “Examples of Complexity OoD” in various domains like robotics, theorem proving, code generation, algorithmic reasoning and document comprehension clearly motivate the need for a complexity axis that is a function of both reasoning length (as opposed to the simplistic solution length in length-OOD) and representational capacity. The distinction between related work -- length OOD and compositional OOD is also sufficient.
1. The breakdown of the proposed “complexity-OOD” into representational and computational axes is of interest to the readers of TMLR, this is a valuable perspective on how to categorize complexity of problems.
1. The connection to LLMs, VLMs (vision language models) is well drawn, with proper citations for various attempts in the community to tackle representational complexity (e.g.: slot attention) and computational complexity (e.g.: variable length reasoning traces).
1. The Kolmogorov Complexity perspective and its practical proxies is a useful way of measuring complexity in the domains mentioned (objects, attributes, relations in image understanding, reasoning steps in mathematical proofs, etc).
1. Section 5.1: “Rethinking Evaluation: Tasks and Benchmarks” is of interest to the community as it makes a valuable observation in the GSM8K and AIME datasets that models having high average accuracy still fail as the number of reasoning steps increases, supporting their core thesis of the need for complexity-OOD benchmarks. The proposed complexity-binned measures would be valuable to include in a future benchmark.
1. Section 5.4 is provides value by pre-emptively identifying issues with System-2 domain problems, specifically the fact that they inherit the failure modes of System-1 primitives while also introducing additional points of failure in the many complex reasoning steps.

## Weaknesses
1. The explanation of duality between System-1 and System-2 tasks (Section 3) is a bit confusing and the benefits of this insight are not explained well.
    * Firstly, the statement “many System-1 tasks transform into System-2 challenges when subjected to complexity-out-of-distribution pressures” is supported by an example of increasing the number + overlap of objects in an object recognition task. This seems to be valid for the representational complexity axis, but an example for the computational steps axis is lacking. The statement “many System-1 tasks transform into System-2” generally seems weak, maybe this should be rephrased as weak duality or more evidence should be provided to support the “many” part of the statement.
    * Secondly, the converse duality of System-2 to System-1 i.e. “Reasoning to Learning”, it is unclear what is meant by the statement “A trained model, however, learns a heuristic function. This heuristic, itself a product of a System-1-like learning process”. How is this heuristic a product of a System-1-learning process? Further, the statement “System-2 act of reasoning is powered by a deeply learned System-1-like intuition that guides its deliberate, step-by-step search.” is also similarly vague.
    * Lastly, how is this duality observation of interest to the broader community? I am not able to understand how this brings about and practical changes to the way users think about or design complexity-aware metrics to reasoning benchmarks.

**Audience:**

Yes

**Audience Explanation:**

As per TMLR’s guidance on this, it seems that this paper more than satisfies this requirement as it provides several valuable insights that will change how future reasoning benchmarks will be developed in many field such as visual understanding, robotics, automated math problem solving, comprehension, etc.

**Broader Impact Concerns:**

No concerns

**Claims And Evidence:**

Yes

**Claims Explanation:**

Mostly yes, except for the duality perspective in Section 3 -- this needs clearer evidence.

**Requested Changes:**

Here are the requested changes:
* (Minor change, will strengthen work, not necessary for securing recommendation for acceptance) Please clarify both sides of the duality insight (Section 3) with sufficient examples and explain how this insight brings value to the community

---

> ### Author Response · Authors · 2026-03-23
>
> We thank the reviewer for their constructive feedback and for validating our core contribution regarding the Complexity OoD framework. We especially appreciate the critique of Section 3, which highlighted a potential ambiguity in our definitions. In response to your comments, we have refined this section to enhance both clarity and practical relevance.
>
> **General Terminology Refinement (System-1/2 Processing vs. Tasks)**
> Before addressing specific points, we wish to highlight a global terminology improvement prompted by your review. We realized that terms like "System-1 Task" or "System-2 Task" could misleadingly imply that tasks possess a fixed cognitive nature. In the revision, we have adopted more precise phrasing, such as "tasks solvable via System-1 processing" and "challenges demanding System-2 processing." This distinction emphasizes that System-1 and System-2 are *modes of processing* rather than inherent task categories, clarifying how a task initially solvable via System-1 can shift to requiring System-2 processing under increased complexity pressure.
>
> **Response to Point 1: Clarifying the S1 $\to$ S2 Transition (Computational Axis)**
> *You correctly noted that while we provided a representational example (visual objects) for the transition from System-1 to System-2, a clear computational example was lacking.*
>
> To address this, we have added two concrete computational examples to Section 3.1:
> * **Pathfinding:** Solving a small, simple maze can often be achieved via visual perception (System-1 pattern recognition). However, as maze size and complexity scale, the same fundamental task forces the agent to adopt a step-by-step search algorithm (System-2).
> * **Arithmetic:** Small multiplication instances (e.g., $12 \times 12$) are typically resolved via memory retrieval (System-1). In contrast, multiplying large numbers (e.g., $342 \times 91$) inherently demands an algorithmic procedure (System-2), even though the rules of the operation remain identical. This illustrates how computational complexity pressure forces a transition from retrieval/perception to deliberate algorithmic reasoning.
>
> **Response to Point 2: Clarifying the "Reasoning to Learning" Duality**
> *You requested clarification on the statement: "A trained model learns a heuristic function... acts of reasoning are powered by a deeply learned System-1-like intuition," specifically asking how this heuristic is a product of a System-1 process.*
>
> To clarify, we conceptualize System-2 processing as the deliberate act of finding a valid sequence of actions to solve a multi-step problem (e.g., the logical steps in a mathematical proof or the sequence of moves in chess).
> * **The "Blind" vs. "Learned" Agent:** An agent with zero prior domain knowledge must rely on brute-force search, blindly exploring all possibilities. Conversely, a trained agent acquires "intuition." Through exposure to correct and incorrect sequences, it learns to recognize patterns in the state space, developing a rapid estimate of which subsequent steps are promising.
> * **The Heuristic as a System-1 Mechanism:** This learned intuition constitutes the "heuristic." Mechanically, it operates as a fast, pattern-matching operation:
>     * *In Large Language Models (LLMs):* Next-token prediction is the atomic System-1 unit, rapidly estimating the most probable continuation based on learned patterns. System-2 "reasoning" is essentially the chaining of these atomic System-1 predictions to navigate a solution space.
>     * *In Reinforcement Learning (e.g., AlphaGo):* Policy and Value Networks fulfill this exact role. They map a complex board state (perception) to a probability distribution over moves or a win-rate estimate. This constitutes fast, learned, System-1 inference.
>
> **Conclusion:** Deliberate, step-by-step search (System-2) is not random; it is heavily guided by learned estimations of optimality. The "heuristic" represents the model's ability to narrow the search space using fast, associative pattern matching (System-1) acquired during training. Without this learned System-1 foundation, efficient System-2 reasoning in complex domains would be computationally intractable.

---

> ### Author Response · Authors · 2026-03-23
>
> **Response to Point 3: Practical Value for Evaluation and Benchmarking**
> *You asked how the duality observation benefits the broader community and impacts benchmark design.*
>
> In response, we have added a new subsection (Section 3.3) detailing these practical implications. The value lies in two necessary shifts in evaluation methodology:
> * **Stress-Testing System-1 Processing:** The first duality provides a framework for stress-testing tasks conventionally solved via System-1 processing. By applying Complexity OoD pressure, we can perform failure analyses to pinpoint exactly when standard perception/memory methods break down. This enables the community to identify where System-2 interventions (algorithmic reasoning or search) are strictly necessary rather than merely beneficial.
> * **The "Heuristic Lens" on System-2 Evaluation:** The second duality dictates that reasoning models should not be evaluated as black boxes. Instead, they must be assessed through the "lens" of the learned heuristic (System-1 intuition) guiding their search. This perspective allows researchers to diagnose whether a reasoning process is masking fundamental learning failures, such as:
>     * *Contamination:* Is the model's "reasoning" merely a regurgitation of memorized patterns?
>     * *OoD Generalization:* Does the heuristic collapse under distribution shifts?
>     * *Spurious Correlations:* Is the search guided by superficial features rather than underlying logical structure?
>     * *Efficiency:* At what computational cost or sample complexity does the heuristic arrive at a solution?
>
> Ultimately, we believe this framework equips the community with concrete metrics to distinguish genuine, robust reasoning from mere pattern-matching on the training distribution.

---

### Review · Reviewer_aytQ · 2026-02-15

**Summary Of Contributions:**

The main contribution of the paper is a framework for defining and measuring reasoning ability of AI systems. The authors build up on the System-1 / System-2 dichotomy from cognitive science, arguing that System-1 tasks have well defined notions of success (OOD generalization) but System-2 lacks a similar framework. The authors propose the notion of Complexity OOD to address this, proposing that reasoning ability should be defined as a model's capacity to generalize to problems whose minimal required solution complexity exceeds anything seen during training. The paper formalizes two notions of Complexity OOD: Representational via Kolmogorov complexity and Computational via conditional Kolmogorov complexity. The paper discusses proxies to these intractable metrics, and reinterpret results on GSM8k by considering complexity as the number of arithmetic operations required to compute the final answer and results on AIME by considering human solution length as a proxy for complexity. The authors then present implication of the framework in the form of practical recommendations for future work.

**Audience:**

No

**Audience Explanation:**

I think in the current state the paper does not fit into the scope of topics at TMLR (https://jmlr.org/tmlr/editorial-policies.html#evaluation). The paper is more of a position paper and does not draw any new connections or function as a comprehensive survey in my opinion.

Aside from fitting in the scope, as I discuss above, a lot of the observations and practical recommendations seem relatively standard even if viewed from a slightly different perspective.

**Claims And Evidence:**

No

**Claims Explanation:**

I have some significant issues with the paper that I detail below. Overall, I believe the claims made in the paper are quite vague, unclear and in some cases not novel. Even taking the claims as is, aside from some analysis of existing results there is not much convincing evidence presented to support the claims.

* Broadly speaking, I do like the question that the paper asks: how can we reliably define and measure reasoning abilities of modern systems? With increasingly sophisticated and strong systems, designing evaluations that accurately reflect the capabilities is a bottleneck, and in the case of reasoning capabilities requires careful thought.
* The paper relies quite heavily on the System-1 / System-2 framework from cognitive science to ground the discussion. While this is somewhat common in the deep learning literature, the distinction is quite thin and somewhat inconsistent within the paper. For instance in Section 2 the authors state that out of distribution generalization is the goal in System 1, but argue that this is insufficient for System-2. This is quite arbitrary, and it is not clear to me why the standard notion of out of distribution is not applicable to the problems the authors discuss in the section (e.g. solving math problems with LLMs). Seems like a somewhat arbitrary distinction.
* The paper uses Kolmogorov complexity to formally define complexity OOD but as the authors note themselves, it is not a practically computable. The authors use some proxies that intuitively correspond to the notions of complexity OOD, but there is no empirical evidence to back the correlation. Additionally, there are no controlled experiments to isolate complexity from confounders. When a model fails on a GSM8K problem, it is unclear if that can be attributed to computational complexity, or things like harder problems also tend to have longer problem statements, more unusual phrasing, less common numerical values, or other distributional shifts. The paper doesn't attempt to disentangle these factors.
* The distinction from compositional and length generalization is again somewhat blurry. Many of the paper's own examples, e.g. Roman numeral conversion, multi-hop VQA, longer arithmetic chains, could equally be described as length or compositional generalization challenges. The authors don't provide a concrete case where Complexity OOD clearly applies but compositional OOD clearly doesn't, or vice versa, in a way that would change how you evaluate or train a model.
* Sections 5.1 through 5.4 propose rethinking evaluation, training, methods, and learning challenges. But the recommendations are broad and largely restate existing research directions: use process-level supervision, build adaptive-depth architectures, address catastrophic forgetting, improve calibration. The paper doesn't prioritize among these directions or provide evidence about which would yield the most improvement and adding little in terms of new insights.
* The paper claims that Complexity OOD generalization "cannot be resolved merely by scaling training data" because one can always construct a harder test instance. But this raises an uncomfortable question: if no finite training regime can ever guarantee Complexity OOD generalization, is the framework setting an impossible standard? And if the answer is that we just want "graceful degradation," then the framework reduces to the already common practice of evaluating scaling behaviour.
* Finally, the arguments in the paper are often handwavy and the writing is quite hard to follow. There is significant repetition and several terms (e.g. Complexity) are used informally in ways incompatible with the formal definition.

**Requested Changes:**

In my view the current manuscript is quite far from polished, and requires significant changes.
* Add precise claims about what the complexity OOD framework contributes to existing perspectives of generalization
* Clarify precisely how compositional and length generalization are differ from complexity OOD.
* Add some evidence to support the proxy metrics being correlated with the formal Kolmogorov complexity definition.
* Additional experiments to isolate the effect of confounders in the evaluations considered in the paper.
* Improving the writing to be clearer and concise.

---

> ### Author Response · Authors · 2026-03-23
>
> We sincerely thank the reviewer for their rigorous review and for pushing us to clarify the empirical and conceptual boundaries of our framework. We appreciate your underlying agreement with our core motivation: designing evaluations that accurately reflect reasoning capabilities is a critical bottleneck in modern AI.
>
> Regarding your concern about the paper fitting within the scope of TMLR, we respectfully refer to the journal's official editorial policies. TMLR explicitly welcomes papers that offer "new approaches for analysis, visualization, and understanding of artificial or biological learning systems," as well as the "development of new analytical frameworks" and "methods for assessing performance." Our manuscript aligns directly with these broader objectives. Rather than presenting a purely philosophical position, we introduce Complexity OoD as a formal analytical framework and diagnostic lens to better understand and evaluate the reasoning limits of modern AI systems. By formalizing this distinction, we aim to provide the community with a more rigorous approach for analyzing when a model genuinely executes an algorithm versus when it relies on memorized shortcuts.
>
> **Response to Point 1:**
> In standard machine learning (System-1 processing), the ability to perform well on unseen data from the training distribution—or a closely related OOD distribution—is the fundamental metric that distinguishes true "learning" from mere "memorization." Our argument is that we critically need a corresponding, structural principle to distinguish true generalization from memorization in tasks requiring System-2 processing.
>
> Standard OOD in perceptual/System-1 tasks typically refers to changes in input features (e.g., Covariate Shift, where $P(X)$ changes). However, the defining challenge for reasoning tasks is not just a shift in surface-level input statistics, but a shift in the solution process and computational depth. A System-2 model faces the persistent danger of acting as a System-1 pattern-matcher—memorizing specific templates, heuristic shortcuts, or bounded solution paths seen during training.
>
> To rigorously prove that a model has learned a generalizable reasoning process (rather than just pattern-matching reasoning traces), it must successfully solve instances requiring deeper, more complex execution paths than anything in its training set. Therefore, our argument is not that standard OOD is irrelevant to System-2, but rather that Complexity OoD is the foundational and most critical OOD setting for System-2 processes.
>
> **Response to Point 2:**
> We acknowledge that Complexity OOD can occasionally intersect with length or compositional generalization; however, they are fundamentally distinct concepts. The necessity for defining Complexity OOD stems from its specific focus on the **solution level** (computational depth and algorithmic process), whereas length and compositional generalization are traditionally defined at the **input level** (surface string length or recombination of input features).
>
> Not every Length or Compositional OOD task qualifies as Complexity OOD. To make this distinction concrete, we have added a new formal example in Section 2.4 of the revised manuscript regarding the evaluation of mathematical expressions:
>
> Consider the task of evaluating arithmetic sequences.
> * **Length OOD:** A model trained on 5-term additions (e.g., $1 + 2 + 3 + 4 + 5$) is tested on a 20-term addition (e.g., $1 + 2 + \dots + 20$). While the input string is significantly longer, the required computational depth per step remains constant, and the algorithmic process is completely flat (sequential accumulation). A longer string does not necessarily equate to a more complex computation.
> * **Complexity OOD:** A model is tested on heavily nested expressions, such as $(1 - (2 - 3) + 4 - (5 - (6 - \dots)))$. Here, the sequence length might be identical to the training data, but the *depth of the computational syntax tree* has increased. The model cannot simply accumulate values left-to-right; it must maintain a deeper state stack.
>
> Similarly, compositional generalization typically refers to the novel recombination of known atomic elements in the input space (e.g., understanding "red sphere" after seeing "red cube" and "blue sphere"). This tests whether a model has disentangled representations, but it does not dictate that the *minimal required computation* to solve the novel composition is strictly deeper or more complex than the training distribution.
>
> Complexity OOD isolates the specific axis of algorithmic depth. By isolating this variable, we can evaluate whether a model has learned a generalizable reasoning process or simply memorized bounded computational graphs. This clarification, along with the nested parentheses example, has been explicitly integrated into Section 2.4 to draw a sharp boundary between these paradigms.

---

> ### Author Response · Authors · 2026-03-23
>
> **Response to Point 4:**
> We thank the reviewer for raising this critical point. To empirically isolate computational complexity from the primary potential confounder—surface-level input length (longer problem statements)—we designed a new set of experiments.
>
> We analyzed the performance of five state-of-the-art reasoning models (Claude 3.7 Sonnet, DeepSeek Reasoning, Gemini 2 Flash, GPT-4o, and GPT-o3-mini) across three distinct mathematical reasoning datasets: AIME, OmniMath, and GSM8K. We computed the Pearson correlation coefficients between:
> 1. Solution Token Count and Question Token Count (to measure the correlation between the variables).
> 2. **Solution Token Count and Model Accuracy** (our proxy for Computational Complexity).
> 3. **Question Token Count and Model Accuracy** (the confounder).
>
> The results of this analysis are summarized in Table 1.
>
> | **Dataset** | **Model** | **Corr(Sol, Quest)** | **Corr(Sol, Accuracy)** | **Corr(Quest, Accuracy)** |
> | :--- | :--- | :--- | :--- | :--- |
> | AIME | Claude 3.7 Sonnet | 0.311 | **-0.390** | -0.187 |
> | | DeepSeek Reasoning | 0.311 | **-0.162** | -0.086 |
> | | Gemini 2 Flash | 0.311 | **-0.332** | -0.260 |
> | | GPT-4o | 0.311 | **-0.201** | -0.092 |
> | | GPT-o3-mini | 0.311 | **-0.279** | -0.130 |
> | OmniMath | Claude 3.7 Sonnet | 0.082 | **-0.126** | -0.033 |
> | | DeepSeek Reasoning | 0.082 | **-0.156** | -0.072 |
> | | Gemini 2 Flash | 0.081 | **-0.191** | -0.072 |
> | | GPT-4o | 0.082 | **-0.103** | -0.008 |
> | | GPT-o3-mini | 0.082 | **-0.150** | -0.115 |
> | GSM8K | Claude 3.7 Sonnet | 0.549 | **-0.072** | -0.051 |
> | | DeepSeek Reasoning | 0.549 | **-0.073** | -0.035 |
> | | Gemini 2 Flash | 0.549 | -0.133 | **-0.137** |
> | | GPT-4o | 0.549 | **-0.267** | -0.161 |
> | | GPT-o3-mini | 0.549 | -0.119 | **-0.139** |
>
> *Table 1: Correlation analysis isolating solution complexity from question length confounders.*
>
> As the data clearly demonstrates, the drop in model accuracy is consistently and significantly more strongly correlated with **Solution Length** (computational complexity) than with **Question Length** (the surface-level confounder).
>
> This is particularly evident in datasets like OmniMath, where the correlation between Question Length and Solution Length is near zero ($\approx 0.08$). In these cases, Question Length has virtually no impact on accuracy (e.g., $r = -0.008$ for GPT-4o), while Solution Length maintains a clear negative correlation ($r = -0.103$). Similarly, in AIME, the negative impact of solution complexity on accuracy is often double that of question length (e.g., Claude 3.7 Sonnet: -0.390 vs -0.187).
>
> These results empirically validate our theoretical framework: while distributional shifts in phrasing or prompt length exist, the degradation in System-2 reasoning performance is fundamentally driven by the depth of the latent computational graph (Complexity OOD) required to reach the answer. We have included this comprehensive empirical analysis in the revised manuscript to address the concern regarding confounders.

---

### Review · Reviewer_LkPT · 2026-02-16

**Summary Of Contributions:**

This position paper proposes Complexity Out-of-Distribution (Complexity OoD) generalization as a unifying framework for defining and evaluating reasoning. The authors argue that reasoning ability should be measured by a model’s capacity to generalize to instances whose minimal solution complexity—either representational or computational—exceeds that of all training examples. They formalize this notion using Kolmogorov complexity and practical proxies (e.g., number of reasoning steps, object counts in a visual scene), and position Complexity OoD as a bridge between System-1 learning and System-2 reasoning. The paper further surveys existing methods (e.g., CoT, RL, adaptive computation) through this lens and advocates for complexity-aware benchmarks and inductive biases.

**Audience:**

Yes

**Audience Explanation:**

Researchers working on reasoning benchmarks, curriculum learning, evaluation methodology, and inductive biases for adaptive computation will likely find the Complexity OoD perspective useful as a conceptual organizing principle. In particular, the emphasis on complexity-conditioned evaluation highlights failure modes that are obscured by aggregate metrics and is directly relevant to current debates around LLM reasoning reliability and contamination. However, interest will be strongest among readers focused on evaluation and diagnostics rather than those seeking concrete algorithmic advances.

**Broader Impact Concerns:**

No significant ethical concerns are apparent.

**Claims And Evidence:**

No

**Claims Explanation:**

As it is, while the paper provides compelling qualitative arguments and illustrative empirical analyses (e.g., complexity-binned accuracy on GSM8K, AIME, Omni-MATH), these results primarily validate the usefulness of complexity-aware reporting, rather than the stronger claim that Complexity OoD provides a sufficient definition of reasoning. In practice, the proposed complexity proxies (operation counts, solution length) are task-specific and rely heavily on human-designed annotations, which, for instance in curriculum learning literature, prove to be weak signals for learning and not necessarily aligned with the optimal learning strategy for artificial neural networks. It remains unclear how the framework applies to unconstrained settings where solution structure is implicit or ill-defined, precisely the regimes in which modern foundation models operate. As a result, the evidence supports the diagnostic value of the framework, but not its generality or sufficiency as a definition of reasoning.

**Requested Changes:**

Major
1. Clarify the scope and limits of “complexity.” The paper should explicitly address how Complexity OoD applies (or fails to apply) in large-scale, weakly constrained domains such as natural language modeling over trillion-token corpora, where solution complexity is neither uniquely defined nor easily separable from data distribution effects. Without this clarification, the framework risks being limited to well-structured tasks (e.g., math, algorithmic reasoning, synthetic benchmarks).
2. Disentangle complexity from curriculum learning. The paper implicitly echoes long-standing challenges in curriculum learning, namely that human-defined difficulty measures often fail to transfer to neural models. A direct discussion of this connection, including why Complexity OoD avoids (or inherits) these failures, is necessary.
3. Tighten the link between the framework and inductive biases. While adaptive computation, external memory, and modularity are discussed extensively, it is not always clear how these mechanisms operationally guarantee or approximate Complexity OoD generalization, rather than simply correlating with improved performance.

Minor but would strengthen the paper:
4. Provide counterexamples or failure cases where complexity proxies break down (e.g., long but trivial solutions, short but highly entangled reasoning).
5. More sharply distinguish Complexity OoD from existing notions of length generalization and compositional generalization with concrete empirical contrasts. While there is a section covering this, I remain a bit unconvinced of the boundaries between the two, especially when complexity is proxied by things like the number of operations, which are highly similar to those found in productivity and length generalization works. For instance, presenting an experiment or task where length/generalization improves but Complexity OoD fails (for instance a model that handles long sequences perfectly but collapses at high reasoning depths, or vice versa) would significantly bolster this argument.
6. Explicitly frame the proposal as a diagnostic and evaluative lens, rather than a complete definition of reasoning, if that is the authors’ intent.

---

> ### Author Response · Authors · 2026-03-23
>
> We sincerely thank the reviewer for their thoughtful, rigorous, and highly constructive feedback. We fully agree with your central assessment: our goal was never to resolve the philosophical debate of "what is reasoning," but rather to provide an operational tool for AI researchers to distinguish genuine algorithmic generalization from memorized shortcuts. You rightly pointed out that our initial phrasing overclaimed by using the word "define." We have embraced this feedback and thoroughly revised the Abstract, Introduction, and Conclusion to explicitly reframe Complexity OoD as a diagnostic framework and a necessary condition for evaluating reasoning, rather than a complete philosophical definition.
>
> Crucially, to directly address your insightful critiques regarding unconstrained domains (Point 1), proxy failure cases (Point 4), and the empirical distinction from length generalization (Point 5), we have added a dedicated new subsection to the manuscript: Section 2.4 ("Scope and Limitations of Complexity Proxies"). We believe this addition significantly strengthens the nuance and rigor of the paper.
>
> Below, we address your specific requested changes point-by-point and outline the corresponding updates made to the manuscript.
>
> **Response to Point 1:**
> We completely agree with the reviewer that Complexity OoD is most naturally and intuitively measured in well-structured, verifiable domains like mathematics, coding, and algorithmic reasoning. However, we argue that the underlying principle of Complexity OoD fundamentally extends to weakly constrained and unstructured domains (such as natural language modeling and NLU), even though measuring it becomes much noisier. In natural language, complexity can be defined in terms of syntactic and semantic depth. For instance:
>
> * Nested Clauses and Syntactic Depth: Consider sentences with complex nested structures (e.g., "The failure of the vote to remove the anti-war representative..."). Comprehending or generating such text requires a model to maintain a larger state in its working memory to resolve long-distance dependencies and subject-verb agreements. A model relying on shallow $n$-gram shortcuts will fail when tested on syntactic depths greater than those seen in training.
> * Semantic Disentanglement: Similarly, processing sentences containing deep metaphors, pragmatics, or implicit multi-hop logical implications requires deeper "semantic processing" and disentanglement compared to literal, direct statement
>
> **Response to Point 2:**
> We completely agree with your observation regarding Curriculum Learning (CL). The CL literature has indeed repeatedly shown that human-defined measures of difficulty do not always align with the optimal learning trajectory for artificial neural networks. However, we would like to clarify two fundamental distinctions between CL and Complexity OoD:
>
> * Evaluation Lens vs. Training Strategy: Curriculum Learning is fundamentally a training strategy (proposing that models should be trained from easy to hard). In contrast, Complexity OoD is purely an evaluation setting and diagnostic lens. We are not claiming that models must be trained in a progressive curriculum; rather, we are stating: "Regardless of the training strategy used, a model truly capable of reasoning must be able to generalize at test time to instances with higher complexity than it saw during training."
> * Objective Computational Complexity vs. Subjective Difficulty: In traditional CL literature, "difficulty" is often subjective, heuristic-based, or model-centric (e.g., what the current model finds "hard" to fit). In Complexity OoD, we focus strictly on objective computational and algorithmic complexity (e.g., the minimum required inference steps, or the depth of a search tree). Particularly in System-2 tasks like mathematics or algorithmic coding, these "human steps" are highly correlated with objective computational steps. Therefore, as an evaluation proxy, this metric is much more rigorous and valid than traditional CL difficulty metrics.
>
> In fact, the Complexity OoD framework perfectly explains why Curriculum Learning often fails in practice. If a model is trained on "easy" examples in a curriculum and learns to solve them using superficial statistical shortcuts rather than true algorithmic rules, it will catastrophically fail when evaluated on the Complexity OoD test set. Therefore, rather than inheriting the flaws of CL, our framework serves as the exact diagnostic tool needed to detect when a model has learned a shortcut instead of a generalizable reasoning process.

---

> ### Author Response · Authors · 2026-03-23
>
> **Response to Point 3:**
> We completely agree with the reviewer that incorporating specific inductive biases (such as Adaptive Computation Time (ACT), external memory, or test-time search) does not absolutely guarantee successful generalization, as the model may still fail during training or optimization. However, we argue that these mechanisms act as Necessary Structural Prerequisites for Complexity OoD generalization.
> The core theoretical bottleneck of standard architectures (like standard vanilla Transformers) is that they possess a fixed computational depth and representational capacity—let us call this capacity $N$ (bounded by the number of layers and fixed context constraints). If a novel test instance requires a minimal sequential reasoning depth of $N+1$ steps to be solved, a standard Transformer theoretically cannot execute it, regardless of how much data it was trained on or how well it was optimized.
> This is exactly where inductive biases come in. Mechanisms like ACT or read/write memory do not merely correlate with better performance; they fundamentally decouple computational depth from parameter count. By allowing a model to dynamically execute as many computational steps as needed (or expanding its state-tracking via memory), these biases provide the model with the theoretical capacity to handle problems requiring $N+1$, $N+10$, or arbitrary algorithmic depths.
> Therefore, while these inductive biases do not guarantee that the model will learn the correct algorithm, they guarantee the capacity to represent a solution of arbitrary complexity. Without them, true Complexity OoD generalization is architecturally impossible.
> We have also incorporated this clarification into Section 5.3 of the revised manuscript to ensure the argument is reflected in the main text.
>
> **Response to Point 4:**
> We fully agree with your observation. We are acutely aware of the limitations of empirical proxies (such as output token count or number of reasoning steps) and recognize that "long" does not always equate to "complex." As requested, we have added a new paragraph titled "Limitations of Proxies" to the manuscript to explicitly discuss where these heuristics break down. We highlight two primary failure modes:
>
> False Positives (High Length, Low Complexity): Sometimes a task is merely "high volume" but logically trivial. For example, asking a model to "Print 'Hello World' 100 times" results in a very long output (high token count), but the underlying generating algorithm is extremely simple (a basic loop with near-zero Kolmogorov complexity). This scenario tests a model's Productivity (the ability to sustain generation without losing positional tracking) rather than its Complexity (the ability to perform deep reasoning).
>
> False Negatives (Low Length, High Complexity): Conversely, some problems require massive computational search or deep "thinking," yet the final verifiable answer is very short. For example, solving a complex logical riddle, cracking a cryptographic hash, or finding a mathematical insight (a "mental spark") might require an immense search tree, but the final proof or answer is only two lines long. In these cases, using the length of the final output as a proxy severely underestimates the true complexity of the problem because it entirely ignores the hidden Search Cost. (This also touches upon how the choice of alphabet/language influences Kolmogorov complexity, where a short string can encode massive computational work).
>
> By explicitly detailing these counterexamples in the revision, we clarify that our empirical proxies are practical approximations, not perfect theoretical measures, and must be chosen carefully based on the domain.

---

> ### Author Response · Authors · 2026-03-23
>
> **Response to Point 5:**
> We deeply appreciate this critique, as it highlights a crucial boundary. We fully acknowledge that in practice, higher complexity often manifests as longer output sequences, which can make "Complexity OoD" and "Length Generalization" appear similar. However, they refer to entirely different failure modes in neural networks.
>
> To sharply distinguish the two, we emphasize that Length Generalization evaluates memory and attention limits, whereas Complexity OoD evaluates algorithmic learning and state-tracking.
>
> To make this boundary concrete, we propose two contrasting scenarios (which we have incorporated into Section 2.1 of the revision):
>
> To empirically isolate algorithmic depth from sequence length, consider evaluating arithmetic expressions.
>
> Flat Expression (Low Depth): (10-1) + (10-2) + (10-3) + (10-4)
> Nested Expression (High Depth): 10 - (10 - (10 - (10 - (10 - 1))))
>
> Both expressions have nearly identical input lengths, identical numbers of operands, and the exact same number of mathematical operations. However, the flat expression has a shallow computational tree (Depth $\approx 2$) and can be solved with shallow parallel routing. The nested expression has a deep computational tree (Depth $= 5$) and strictly requires sequential state-tracking.
> When models fail on the nested expression but succeed on the flat one, the failure is purely a failure of Complexity OoD, not Length Generalization. The model is sensitive to the depth of the reasoning tree (the solution representation), not the raw length of the input or output string.
>
> **Response to Point 6:**
> We are in complete agreement with the reviewer on this point. Our objective with this paper was never to resolve the grand philosophical debate of "what constitutes thinking," nor to provide a rigid, all-encompassing definition of reasoning for cognitive science.
>
> Rather, our goal is highly practical: we aim to provide an operational metric for AI engineers and researchers. As models become more sophisticated, it becomes increasingly difficult to distinguish whether a model is genuinely executing a learned algorithm ("reasoning") or merely relying on sophisticated pattern matching and shortcut learning ("memorizing"). We offer Complexity OoD strictly as a diagnostic lens to make this distinction. It gives researchers a concrete, evaluative tool to test if their model is truly reasoning, or merely mimicking reasoning on in-distribution data.
>
> As mentioned in our general response, we have thoroughly revised the Abstract, Introduction, and Conclusion to remove the claim of "defining" reasoning, explicitly reframing the paper around this diagnostic and evaluative perspective.

---

### Decision · Action_Editor_uVnU · 2026-06-14

**Recommendation:** Accept with minor revision

**Additional Comments:**

I think the idea of using complexity as a guide for LLM diagnostics is intuitive, and certainly not in and of itself completely novel. However, the paper's characterization of representational vs computational complexity, insistence on computational depth as a measure of complexity, and ideas like CPC, breakdown point, and degradation slope give it enough concrete leverage to be an interesting contribution.

This paper doesn't offer an immediate tool or method. The diagnostic framing requires that one first develop a proxy metric for complexity, which it itself the more challenging problem. I think that the paper would be strongest if it offered more prescriptive guidance on which proxies to choose. The correlation experiment is a start toward this.

The boundary between weakly and strongly constrained domains should be highlighted, but I think this is a future consideration. That is, if we had a rigorous, general, and computationally tractable measure of complexity in domains like math and coding, then I think the question of how to generalize to unstructured text in NLP domains would be a good problem to have.

For the paper, concretely: please make sure to address the reviewer concerns, and most importantly: fold in the new results, and the example around computational depth vs token length, as they strengthen the paper's arguments greatly.

I do have one lingering question (not evaluative for acceptance, but more of a high-level thought): computational depth does not necessarily make a problem more complex. In many math problems, a valid solution from one perspective might have a cumbersome computational complexity, but then applying a coordinate change makes the solution depth trivial. This doesn't matter from a Kolmogorov complexity perspective, since we would be looking at the minimum complexity among all possible solutions, but practically it means that computational depth can sometimes be misleading.

It is noted that a de-anonymized version was uploaded during the revision cycle. Given the extenuating regional circumstances and internet disruptions communicated by the authors, and because it did not bias the review process, I believe this should not affect acceptance.

**Audience:**

Yes

**Audience Explanation:**

The paper explores interesting philosophical and practical questions from the perspective of a diagnostic framework for LLMs. This is a massive area, and approaching it from a computational and complexity theory perspective is a nice perspective.

The question is whether it falls on the side of philosophical treatise (less interesting for TMLR) or immediate and practical tool (more interesting). It clearly falls somewhere in between, but in my opinion there is enough of an attempt to make it concrete that it provides a n interesting foundation, as opposed to a rigorous method. To that end, I think this criterion is also satisfied.

**Claims And Evidence:**

Yes

**Claims Explanation:**

The paper is more of a conceptual framework than a ready-to-deploy, sharp diagnostic. It provides a roadmap for the development of future diagnostics.

The main "metrics" are based on uncomputable quantities (Kolmogorov complexity) and therefore the utility of the framework rests on the ability to prescribe proxy measures.

However, given a suitable proxy (which is obviously a major challenge), the complexity-performance curve, breakdown point, and degradation slope are well-motivated.

The authors' post-rebuttal correlation analysis on mathematical datasets provides strong empirical support for their framework. By proving that accuracy drops are much more strongly correlated with the computational solution length than with question length, they successfully isolate reasoning depth from sequence-length confounders.

There is also a question about applying this framework to large-scale, weakly constrained domains (Reviewer LkPT), but I think that this is further ahead, and the question of how to apply this to strongly constrained domains (which are still highly relevant for LLM evals) should be fully explored first.

Overall, I think the claims are sufficiently supported by the evidence provided. Not completely justified, but sufficiently.

---

> ### Author Response · Authors · 2026-06-17
> **Camera-Ready Revision: Summary of Changes**
>
> Dear Professor Swersky,
>
> Thank you very much for your careful and thorough reading of our paper, and for your thoughtful decision and guidance. We are especially grateful for your constructive high-level comments, which helped us further sharpen the contribution. We have prepared a revised camera-ready version that incorporates the changes you requested. Below we summarize the specific modifications and where they appear.
>
> 1. Folding in the new correlation results. As you recommended, we have integrated the post-rebuttal correlation analysis directly into the main text. Within Section 5.1 (Rethinking Evaluation), following the complexity-binned analysis of GSM8K, AIME, and Omni-MATH, we now include the full correlation table across five models and three datasets, together with a discussion of how it isolates reasoning depth from sequence-length confounders. We explicitly note that accuracy degradation correlates more strongly with solution length (our computational-complexity proxy) than with question length, with the cleanest effect on Omni-MATH where the two length axes are nearly decoupled.
>
> 2. Folding in the computational depth vs. token length example. We have moved the flat-versus-nested arithmetic example into the main conceptual text. It now appears in Section 2.1, within the discussion distinguishing Complexity OoD from length and compositional generalization, where it provides a concrete case in which length generalization and Complexity OoD dissociate. We connect this directly to proxy design in Section 2.4, where the maximum depth of the computational syntax tree, rather than raw token count, emerges as the faithful complexity proxy.
>
> 3. Distinguishing Complexity OoD from curriculum learning. Following Reviewer LkPT's request that this connection be discussed in the paper itself (rather than only in our rebuttal), we added a dedicated discussion to Section 2.4. We clarify that the well-known critique of curriculum learning, namely that human-defined difficulty measures often fail to align with the optimal learning trajectory of neural networks, targets the use of such measures as a training signal, and therefore does not transfer to our setting: a Complexity OoD proxy is an evaluation and diagnostic instrument, not a difficulty signal for ordering training.
>
> We have also addressed the remaining reviewer concerns and reflected the corresponding clarifications in the manuscript.
>
> Finally, we would like to address the matter of the de-anonymized version uploaded during the revision cycle. We sincerely apologize for this lapse, which we have now realized ourselves. As you kindly acknowledged in your decision, the revision was prepared amid extenuating regional circumstances and internet disruptions: during this period our country was in a state of war, with ongoing military attacks, and for most of the time the internet was effectively cut off, returning only sporadically at unpredictable moments and, even then, extremely slow and unreliable. Under these conditions we were not as careful as we should have been, and the de-anonymized file was uploaded inadvertently. We are deeply grateful for your understanding on this point, and we apologize again for the oversight.
>
> Thank you once again for your time, care, and support throughout this process.
>
> Best regards,
> The Authors

---

> ### Author Response · Authors · 2026-06-17
> **Response to Your Lingering Question on Computational Depth**
>
> We would also like to return to the lingering high-level question you raised in your decision: namely, that computational depth does not necessarily make a problem more complex, since a problem with a cumbersome solution from one perspective may become trivial in depth after a change of coordinates. We found this a genuinely interesting question, and thinking it through was rewarding for us as well. We believe the phenomenon you point to is real, and that it actually fits neatly within the framework, in three complementary ways.
>
> First, our formal definition already anticipates exactly this. Because $K(y \mid x)$ is a minimum over all valid solutions, the depth of any one particular solution is only an upper bound on the true complexity. A clever change of representation is precisely the act of finding a shorter, shallower program; if such a transformation exists, then by definition the problem has low complexity. So a cumbersome solution that looks deep is not evidence that the problem is intrinsically complex, only that the solution we happened to inspect is suboptimal. This is exactly why, in Section 2.4, we distinguish the depth of a single produced trace from the intrinsic complexity of the problem, and why we caution that a proxy based on one human- or model-generated solution can overestimate the true difficulty.
>
> Second, complexity is always defined relative to the set of primitives (the abstraction level) available to the solver, and the change of coordinates is not free. To exploit it, the model must actually possess that representation, whether in its primitives or in what its training distribution affords. If the helpful transformation is available to the model, then the problem genuinely is easy for that model; if it is not, the model is forced down the deep path, and the problem genuinely is hard for it. So rather than undermining the proxy, this relativity is part of what gives the diagnostic its meaning: it measures complexity with respect to the representational vocabulary the model can actually deploy.
>
> Third, and we think most interestingly, the ability to discover such a change of coordinates is itself a hallmark of reasoning. A transformation either applies only to a single instance (a one-off trick), in which case it does not help a model generalize across a growing family of harder instances, or it systematically simplifies a whole family, in which case it is precisely a reusable algorithm or primitive that the model has learned, exactly the kind of System-2 generalization we want to measure. In other words, finding the trivializing coordinate change is not a way of escaping the complexity test; it is the very capability the test is designed to reveal. A model that reliably finds these shortcuts is demonstrating the algorithmic understanding we are after, while a model that cannot is left executing the deep path and will be exposed under Complexity OoD pressure.
>
> We are grateful for raising this point, as it helped us articulate more clearly why single-trace depth is an upper bound rather than the faithful measure, and why the framework is robust to (indeed, partly motivated by) exactly this kind of representational shortcut.